# Decomposing heterogeneous dynamical systems with graph neural networks

## Abstract

Natural physical, chemical, and biological dynamical systems are often complex, with heterogeneous components interacting in diverse ways. We show how simple graph neural networks can be designed to jointly learn the interaction rules and the latent heterogeneity from observable dynamics. The learned latent heterogeneity and dynamics can be used to virtually decompose the complex system which is necessary to infer and parameterize the underlying governing equations. We tested the approach with simulation experiments of interacting moving particles, vector fields, and signaling networks. While our current aim is to better understand and validate the approach with simulated data, we anticipate it to become a generally applicable tool to uncover the governing rules underlying complex dynamics observed in nature.

## 1 Introduction

Many natural phenomena can be modeled (or reasonably approximated) as dynamical systems of discrete particles or finite elements that change their state based on some internal program, external forces, and interactions with other particles or elements. Well known historic examples that use such models for forward simulation are cinematographic applications (Reeves, 1983), Reynolds's boid flocking behavior model (1987), atmospheric flow (Takle & Russell, 1988), and fluid dynamics (Miller & Pearce, 1989).

Particle systems and finite element methods can also be used to uncover the underlying dynamics from observations. If the governing equations of the dynamics are known, it is generally possible to recover the underlying properties of objects from noisy and/or incomplete data by iterative optimization (e.g. Kalman filter; Shakhtarin, 2006). Conversely, if the properties of objects are known, it is possible to determine the governing equations with compressed sensing (Brunton et al., 2016), equations-based approaches (Stepaniants et al., 2023) or machine learning techniques, including graph neural networks (GNN; Battaglia et al., 2016; Cranmer et al., 2020; Sanchez-Gonzalez et al., 2020; Prakash & Tucker, 2022). Recent methods jointly optimize the governing equations and their parameterization (Long et al., 2018; Huang et al., 2020; Lu et al., 2022; Course & Nair, 2023), yet heterogeneity of objects and interactions is either not considered or provided as input.

Zhao et al. (2023) add a learnable convection term to partial differential equation (PDE)-GNNs to account for behavior between heterogeneous particles, leading to improved performance on several classification benchmarks. Interestingly, this term has no access to the particle features but only their relative differences, which limits its ability to learn particle-type specific interaction rules.

In their work on rediscovering orbital mechanics in the solar system, Lemos et al. (2023) explicitly model the mass of orbital bodies as a learnable parameter. They use GNNs to learn how to predict the observed behavior and the latent property, and combine this purely bottom-up approach with symbolic regression to infer and parameterize a governing equation. With this approach, they are able to uncover Newton's law of gravity and the unobserved masses of orbital bodies from location over time data alone.

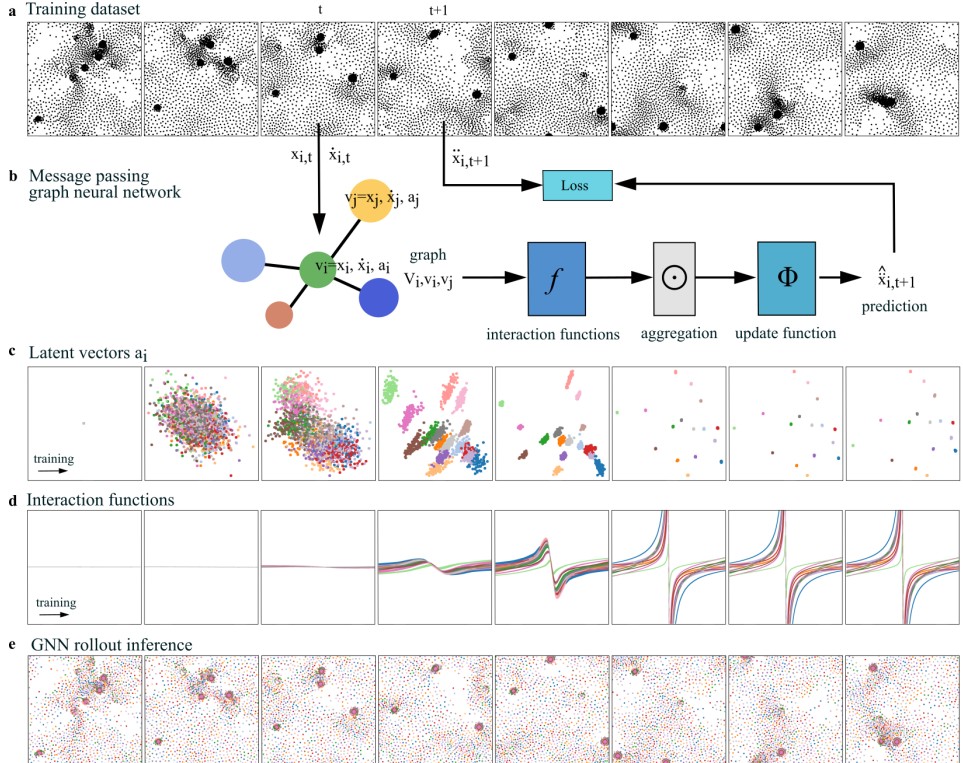

Figure 1: Outline of the GNN method for modeling heterogeneous dynamical system from data. The training dataset (**a**, boids example) is converted into (**b**) a graph time-series (node features $v_i$, connectivity $V_i$) to be processed by a message passing GNN. In all our simulations, except for the signal passing network, the length of the time series used for training is 1, aggregating time variant properties such as velocity in $v_i$. Each particle is represented as a node $i$ that receives messages from connected nodes $j \in V_i$ processed by a pairwise message passing function $f$. These messages are aggregated by a function $\bigodot$ and then used by an update function $\Phi$ to modify the node states. Either of the functions $f$, $\bigodot$, or $\Phi$ can be hard-coded or a learnable neural network. In addition to observable particle properties (here the positions $x_i$ and velocity $\dot{x}_i$), the functions have access to a learnable latent vector $a_i$. During training, the latent vectors for each node and the learnable functions are jointly optimized (**c**, **d**) to predict how particle states evolve over time (**e**). The trained latent embedding reveals the structure of the underlying heterogeneity and can be used to decompose and further analyze the dynamical system.

## 1.1 Contribution

We expand the work by Lemos et al. (2023) to predict and decompose heterogeneous dynamical systems that are governed by arbitrary latent properties. We train GNNs to reproduce the observable dynamics of complex systems. We train only one shared function approximator for all interactions and updates, respectively, that is parameterized by the observable particle properties and a low-dimensional learnable embedding of the latent properties for each node (see Figure 1). In systems with discrete classes of particles, the learned embedding of all nodes reveals the classes as clusters and allows to virtually decompose the system. This is a necessary step to infer and parameterize the underlying governing equations. In systems with continuous properties, the learned embedding reveals the underlying manifold and allows to estimate the corresponding parameters.

For a diverse set of simulations, we can learn to reproduce the complex dynamics, uncover and visualize the structure of the underlying heterogeneity, and parameterize symbolic top-down hypotheses of the rules governing the dynamics. In the simplest cases, the interaction functions were automatically retrieved with symbolic regression.

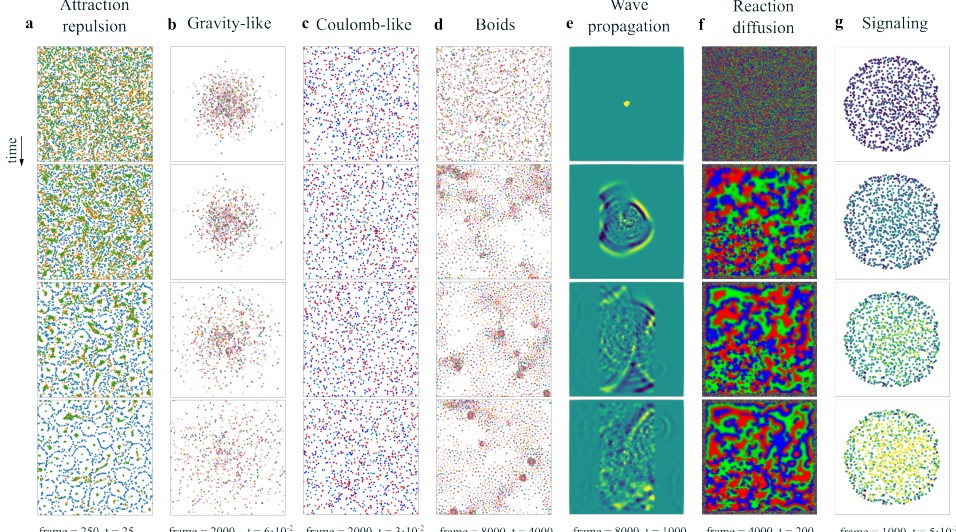

Figure 2: Simulations of dynamical systems. (**a**) Attraction-repulsion, 4,800 particles, 3 particle types. (**b**) Gravity-like, 960 particles, 16 different masses. (**c**) Coulomb-like, 960 particles, 3 different charges. (**d**) Boids, 1792 particles, 16 types. (**e**) Wave-propagation over a mesh of $10^4$ nodes with variable propagation-coefficients. (**f**) Reaction-diffusion propagation over a mesh of $10^4$ nodes with variable diffusion-coefficients. (**g**) Signaling network, 986 nodes, 17,865 edges, 2 types of nodes. The underlying equations are detailed in Supplementary Table 1.

## 2 METHODS

### 2.1 SIMULATION OF DYNAMICAL SYSTEMS

We created a diverse set of quasi-physical dynamical particle systems with heterogeneous latent properties of individual particles (see Figure 2 and Videos). In all simulations, particles interact with a limited neighborhood of other particles. They receive messages from connected particles that encode some of their properties, integrate these messages, and use the result to update their own properties. This update is either the first or second derivative over time of their position or other dynamical properties.

First, we created simulations of moving particles whose motion is the result of complex interactions with other particles (Lagrangian systems, see Figure 2a–d). Then, we simulated vector-fields with diffusion-like signal propagation between stationary particles (Eulerian systems, see Figure 2e, f). Some particles follow an exclusively internal program defined by a sequence of hidden states. Finally, we created complex spatio-temporal signaling networks (Hens et al., 2019; see Figure 2g).

All simulations generate indexed time series by updating parts of all particle states $x_i$ using explicit or semi-implicit Euler integration

$$\dot{x}_i \leftarrow \dot{x}_i + \Delta t \, \ddot{x}_i, \quad x_i \leftarrow x_i + \Delta t \, \dot{x}_i. \tag{1}$$

The vector $x_i$ stands for the position of particles in moving dynamical systems, or for other dynamical properties in the vector-field and network simulations. The details of these simulations are listed in Supplementary Table 1.

### 2.2 GRAPH NEURAL NETWORKS

Figure 1 depicts the components of the GNNs to model dynamical particle systems, and how we train them to predict their dynamical behavior and to reveal the structure of the underlying heterogeneity. A graph $G = \{V, E\}$ consists of a set of nodes $V = \{1, \ldots, n\}$ and edges $E \subseteq V \times V$ with node and edge features denoted by $v_i$ and $e_{ij}$ for $i, j \in V$, respectively. A message passing GNN updates

node features by a local aggregation rule (Battaglia et al., 2016; Gilmer et al., 2017)

$$v_i \leftarrow \Phi\Big(v_i, \bigodot_{j \in V_i} f(e_{ij}, v_i, v_j)\Big), \tag{2}$$

where $V_i := \{j : (i,j) \in E\}$ is the set of all neighbors of node $i$, $\Phi$ is the update function, $\bigodot$ is the aggregation function, and $f$ is the message passing function. To model a dynamical particle system, $\Phi$, $\bigodot$, and $f$ can be used to represent the time evolution of node states according to pairwise and node-local interactions. The node features $v_i$ include the dynamical node states $\boldsymbol{x}_i$ ($\boldsymbol{x}_i \in \mathbb{R}^d$). In models with moving particles, $\boldsymbol{x}_i$ is the position of the particles. In models with stationary particles it stands for their dynamical properties. With this framework, we can model arbitrary dynamical particle systems by using particles as nodes and designing an appropriate neighborhood, node and edge features, as well as update, aggregation, and message passing functions. Either $\Phi$, $\bigodot$, or $f$ can be arbitrary differentiable functions, which includes fully learnable deep neural networks. In our experiments, we use multi-layer perceptrons (MLPs) for such learnable functions, and typically, only $f$ or parts of $\Phi$ are fully learnable at a time. The aggregation function $\bigodot_i$ is either the sum or the average of the outputs of all $f_{ij}$. The inputs to these functions are application specific subsets of the node features, such as the relative position between the particles, $\boldsymbol{x}_j - \boldsymbol{x}_i$, the distance between particles $d_{ij}$ or the velocity of the particles $\dot{\boldsymbol{x}}_i$. The latent heterogeneity of the particles is encoded by a two-dimensional learnable embedding $\boldsymbol{a}_i$ that is part of the node features. These learnable embeddings parameterize either $\Phi$, $\bigodot$, or $f$ as appropriate. For all our experiments with one to four-dimensional latent parameterers, two- or more dimensional embeddings generated similar results. We therefore chose two dimensions, because they are easy to visualize and interpret. Experiments with one-dimensional embeddings get often stuck in local minima. We expect that higher-dimensional latent parameter spaces that are less sparse would require higher-dimensional embeddings.

The design choices for neighborhood, learnable functions, and their parameters are important to define what the GNN can learn about the dynamical system. If either of the learnable functions has access to the absolute position $\boldsymbol{x}_i$ of particle node $i$ and the time index $t$, then this function can learn the behavior of the particle as a fully independent internal program. This is sometimes desired, e.g. if we want to learn the behavior of an unobserved force-field that impacts the behavior of observable dynamical particles (see Figure 4). If the learnable interaction function $f$ has only access to local relative offsets, velocities, and distances, then it has to learn to reproduce the dynamics based on these local cues (see Figure 3). We found that the networks learn to ignore redundant input parameters that are irrelevant for the task, e.g. networks that learn to infer gravitational forces from relative positions learn to ignore velocities or accelerations, even if they have access to those derivatives. Please see Supplementary Table 2 for a full description of the GNN models used for the various simulation experiments.

During training, the learnable parameters of $\Phi$, $\bigodot$, and $f$, including the embedding $\boldsymbol{a}_i$ of all nodes $i \in V$ are optimized to predict a single time-step or a short time-series. Since we use explicit or semi-implicit Euler integration to update the dynamical properties of all particles (see Equation 1), we predict either the first or second order derivative of those properties and specify the optimization loss over those derivatives

$$L_{\dot{\boldsymbol{x}}} = \sum_{i=1}^{n} \|\widehat{\dot{\boldsymbol{x}}}_i - \dot{\boldsymbol{x}}_i\|^2 \quad \text{and} \quad L_{\ddot{\boldsymbol{x}}} = \sum_{i=1}^{n} \|\widehat{\ddot{\boldsymbol{x}}}_i - \ddot{\boldsymbol{x}}_i\|^2. \tag{3}$$

We implemented the GNNs using the PyTorch Geometric library (Fey & Lenssen, 2019). For GNN optimization we used AdamUniform gradient descent, with a learning rate of $10^{-3}$, and batch size of 8. For models with rotation invariant behaviors, we augmented the training data with 200 random rotations. Each GNN was trained over 20 epochs, with each epoch covering all time-points of the respective training series (between 250 and 8000). All experiments were performed on a Colfax ProEdge SX4800 workstation with an Intel Xeon Platinum 8362 CPU, 512 GB RAM, and two NVIDIA RTX A6000 GPUs with 48 GB memory each, using Ubuntu 22.04.

## 2.3 RESULTS

### 2.3.1 ATTRACTION-REPULSION

We simulated a dynamical system of moving particles whose velocity is the result of aggregated pairwise attraction-repulsion towards other particles within a local neighborhood (see Figure 2a

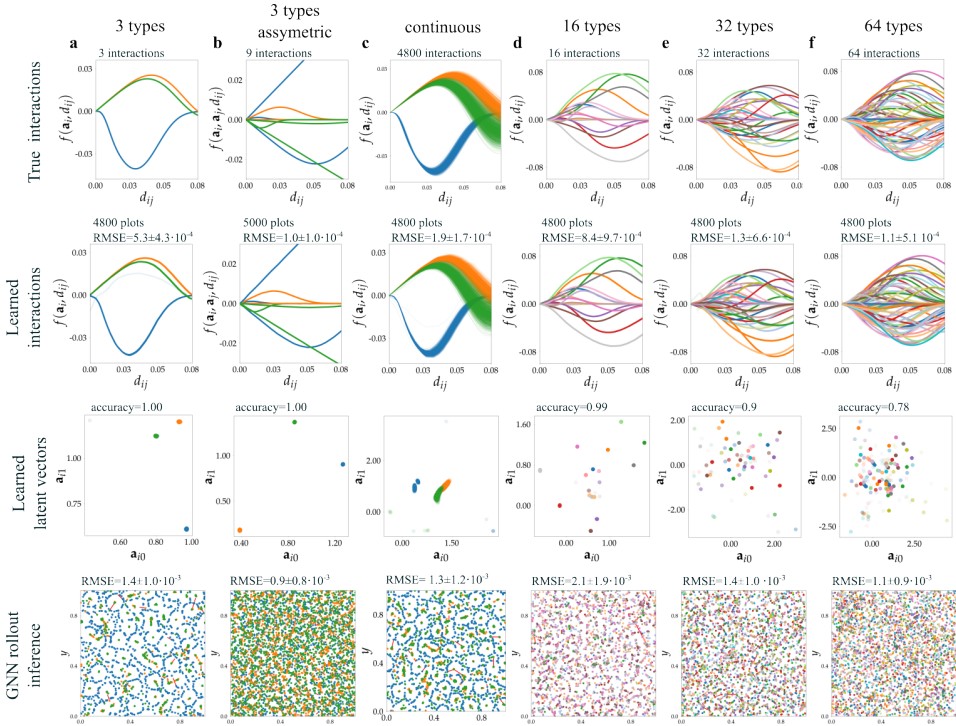

Figure 3: Experiments with the attraction-repulsion model. Row 1 shows the projection of the true interaction functions $f$ speed over distance. Row 2 shows the projection of the learned interaction functions $f$. Row 3 shows the learned latent vectors $\boldsymbol{a}_i$ of all particles. Particle classification is obtained with hierarchical clustering (see Appendix A.2). Row 4 shows the last frame of rollout inference of the trained GNN on a validation series of 250 time-steps. Deviation from ground truth shown by red segments. RMSE measured between true positions and GNNs inferences are given ($\boldsymbol{x}_i \in [0, 1)^2$, 4,800 particles, 250 time-steps). (**a**, **b**) Three particle types. (**c**) Continuous particle parameters. (**d**, **e**, **f**) 16, 32, 64 particle types. In (**b**), the interaction functions asymmetrically depend on the types of both particles, in all other experiments, they depend only on the type of the receiving particle. Colors indicate the true particle types.

and Appendix A.1.1, 4,800 particles, 250 time-steps). The velocity incurred by pairwise attraction-repulsion depends on the relative position of the other particle and the type of the receiving particle (see Supplementary Table 1).

For a simulation with three particle types, we visualized the training progress of the learned embedding vectors $\boldsymbol{a}_i$ for all particles $i$, and the learned interactions $f_{ij}$ (see Supplementary Figure 1 and Video 1). For all experiments, we show an informative projection of the learned pairwise interactions (here, speed as a function of distance).

Initialized at 1, the vectors $\boldsymbol{a}_i$ separate and eventually converge to a clustered embedding that indicates the three distinct node types (and one outlier). The corresponding interaction functions capture the simulated attraction-repulsion rule increasingly well, and also recover that there are three distinct groups (and one outlier).

With limited data, it is not always possible to learn a perfect clustering. We therefore applied a heuristic that can be used when the structure of the learned embedding suggests that there is a small number of distinct groups. Every 5 out of 20 epochs, we performed hierarchical clustering on a UMAP projection (McInnes et al., 2018) of the learned interaction function profiles (see Appendix A.2). We then replace, for each particle, the learned latent vectors $\boldsymbol{a}_i$ by the median of the closest cluster, independently estimated for each embedding dimension. With this re-initialization, we continue training the GNN (see Video 1). This bootstrapping helped us to identify the correct number of types in datasets with limited training data.

Figure 3a shows the training results for this model with three particle types whose interactions consider only the receiving type. Hierarchical clustering of the learned latent vectors $\boldsymbol{a}_i$ recovered the particle types with a classification accuracy of $1.00$ (4,800 particles, 3 types, 1 outlier). The root mean squared error (RMSE) between the learned and true interaction function profiles is $5.3 \pm 4.3 \cdot 10^{-4}$. We also measured the accuracy of rollout inference on a validation dataset (see Video 2). The RMSE between true and inferred particle positions was $1.4 \pm 1.0 \cdot 10^{-3}$ ($\boldsymbol{x}_i \in [0, 1)^2$, 4,800 particles, 250 time-steps). We show quantitatively that the trained GNN generalizes well if we change the number of particles and the initial conditions (see Supplementary Figure 2). Most importantly, the optimized GNN can be used to virtually de- or re-compose the dynamic particle system from the identified sub-domains (see Supplementary Figure 3). This ability will be particularly important to understand the behavior of heterogeneous dynamical systems in biology that can only be observed in their mixed natural configuration.

Figure 3b shows the results for a system with three particle types that interact asymmetrically depending on the types of both particles. The GNN learns all nine modes of interaction (RMSE = $1.0 \pm 1.0 \cdot 10^{-4}$) and permits classification of the underlying particle types based on the learned latent vectors $\boldsymbol{a}_i$ with an accuracy of $1.00$.

Figure 3c shows how well the model was able to recover continuous heterogeneity. We added Gaussian noise to the parameters of the attraction-repulsion rules used in the first experiment, leading to slightly different behavior of each particle in the simulation. The GNN was not able to recover the behavior of this dynamical system well when we used 250 time-steps as in the previous experiments. However, with 1,000 time-steps, it excellently reproduced the behavior of the system with a validation rollout RMSE of $1.3 \pm 1.2 \cdot 10^{-3}$, and an interaction function RMSE of $1.9 \pm 1.7 \cdot 10^{-4}$.

Figure 3d–f shows the results of experiments with more particle types (16, 32, 64). 16 types are identified perfectly from a training series with 500 time-steps (accuracy= $0.99$, interaction function RMSE= $8.4 \pm 9.7 \cdot 10^{-5}$), but the performance begins to degrade with 32 types (accuracy= $0.9$) and even more so with 64 particle types (accuracy= $0.78$), even though we increased the length of the training series to 1,000 time-steps. We believe that there were simply not enough representative samples for all possible interactions in the training data, because we did not increase the number of particles nor the field of view of the simulation.

We then tested how robust the approach is when the data is corrupted. We used a modified loss $L_{\dot{\boldsymbol{x}}} = \sum_i^n \|\widehat{\dot{\boldsymbol{x}}}_i - \dot{\boldsymbol{x}}_i(1 + \varepsilon)\|^2$ where $\varepsilon$ is a random vector drawn from a Gaussian distribution $\varepsilon \sim \mathcal{N}(0, \sigma^2)$. Corrupting the training with noise has limited effect up to $\sigma = 0.5$ (see Supplementary Figure 4). The learned latent embedding spread out more, but the interaction functions were correctly learned and could be used for clustering after UMAP projection of the profiles. Hiding parts of the data had a more severe effect. Removing 10% of the particles degraded accuracy to $0.67$ and ten-fold increased the interaction function RMSE (see Supplementary Figure 5b). This effect can be partially addressed by adding random 'ghost-particles' to the system. While the trained GNN was not able to recover the correct position of the missing particles, the presence of 'ghost-particles' during training recovered performance for 30% missing particles (interaction function RMSE = $3.2 \pm 2.1 \cdot 10^{-4}$, accuracy = $0.98$), but started degenerating at 30% (see Supplementary Figure 5c, d).

Finally, we added a hidden field that modulates the behavior of the observable dynamics (see Appendix A.1.2, 4,800 particles, 3 types). The field consists of $10^4$ stationary particles with a given latent coefficient $b_i$. These stationary particles interact with the moving particles through the same attraction-repulsion but weighted by $b_i$. During training of the GNN, the values of $b_i$ are modeled by a coordinate-based MLP. Supplementary Figure 6 shows that it is possible to learn the hidden field together with the particle interaction rules from observations of the dynamic particles alone (see Video 3). We then made the hidden field $b_i$ time-dependent and added the time index $t$ as a parameter to the coordinate-based MLP. Similarly to the stationary hidden field $b_i$, Figure 4 and Video 4 show that the GNN was able to recover both the time-dependent hidden field $b_i(t)$ and the particle interaction rules from observations of the dynamic particles alone.

### 2.3.2 GRAVITY-LIKE

In the gravity-like simulation (see Figure 2b and Appendix A.1.1), particles within a reasonable distance (this is not physical, but reduces complexity) attract each other based on Newton's law of

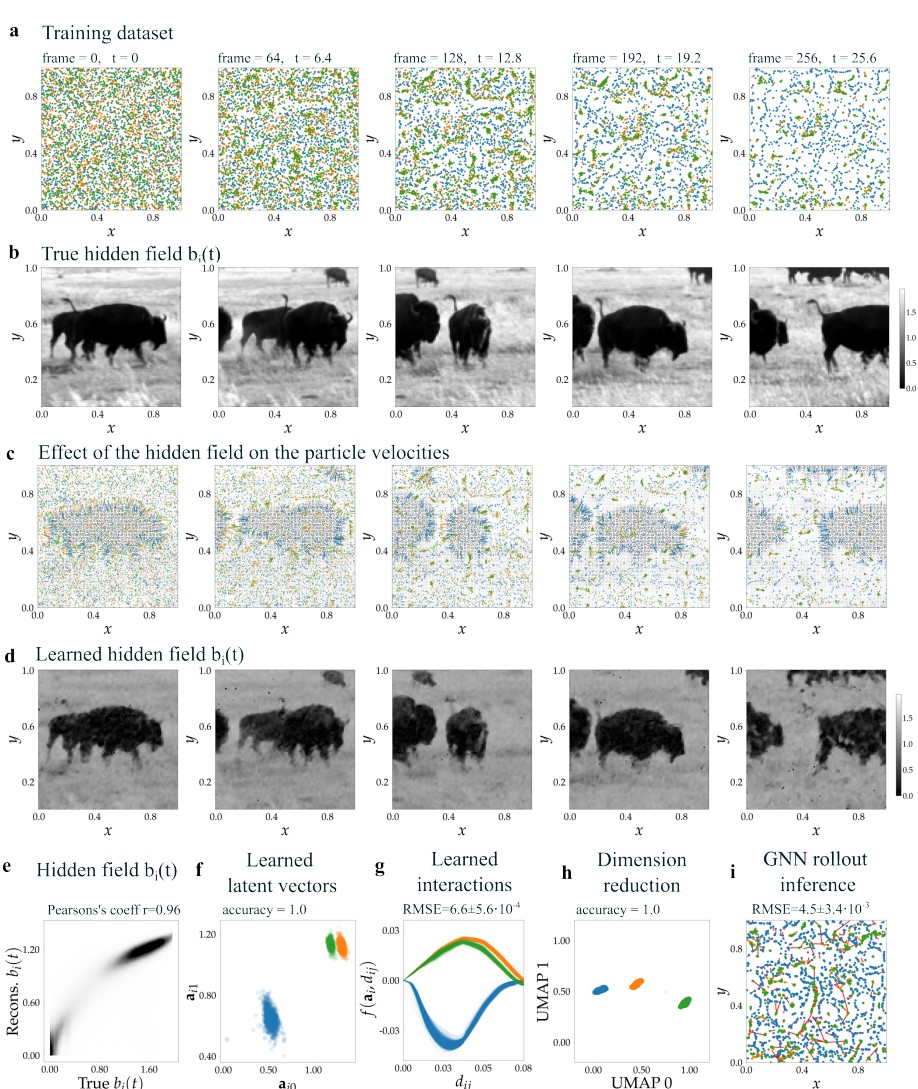

Figure 4: Experiment with the attraction-repulsion model whose particles interact with a hidden dynamical field. (**a**) Snapshots of the training series with 256 time-steps, 4,800 particles, 3 particle types (colors). (**b**) A hidden dynamical field is simulated by $10^4$ stationary particles with time-dependent states $b_i(t)$. (**c**) Small colored arrows depict the velocities of all moving particles as induced by the hidden field (grey dots). (**d**) Hidden field $b_i(t)$ learned by the trained GNN. Structural similarity index (SSIM) measured between ground truth and learned hidden field is $0.46 \pm 0.03$ (256 frames). (**e**) Comparison between learned and true hidden field $b_i(t)$ ($2.6 \cdot 10^6$ points). Pearson correlation coefficient indicates a positive correlation ($r = 0.96$, $p < 10^{-5}$). (**f**) Learned latent vectors $\boldsymbol{a}_i$ for all moving particles. Colors indicate the true particle type. Hierarchical clustering of the learned latent vectors $\boldsymbol{a}_i$ allows to classify the moving particles with an accuracy of 1.00. (**g**) Projection of the learned interaction functions $f_{ij}$ as speed over distance. (**h**) Hierarchical clustering of the UMAP projection of the profiles shown in (**g**) allows to classify the moving particles with an accuracy of 1.00. (**i**) Rollout inference of the trained GNN on 256 time-steps. Deviation from ground truth is shown by red segments. RMSE$= 4.5 \pm 3.4 \cdot 10^{-3}$ ($\boldsymbol{x}_i \in [0, 1)^2$).

universal gravitation which depends on their observable relative position and their latent masses (see Supplementary Table 1).

Supplementary Figure 8a, b shows the results of the GNN trained on two series of 2,000 time-steps with 960 particles of 16 different masses and a continuous distribution of masses, respectively (see Video 5). The GNNs trained with these datasets do not yield precise rollout inference owing to error accumulation (RMSE $\sim 1.0$, Supplementary Figure 7 and Video 6). However the resulting dynamics are qualitatively indistinguishable from the ground truth. This is consistent with a Sinkhorn divergence of $1.1 \cdot 10^{-2}$ between true and inferred distributions (calculated with the GeomLoss library, Feydy et al., 2018). As described by Lemos et al. (2023), we were able to automatically infer and parameterize the symbolic interaction function using the PySR package (Cranmer, 2023). Symbolic regression recovered the $m_i/d_{ij}^2$ power laws (see Supplementary Figure 8a,b) and the 16 distinct masses (slope= 1.01, $R^2 = 1.00$) as well as the continuous distribution of masses (slope= 1.00, $R^2 = 1.00$, 1 outlier).

Corrupting the training with noise has limited effect up to $\sigma = 0.4$ (see Supplementary Figure 9a, b). Removing particles from the training data degraded the results severely (Supplementary Figure 10a, b). Naïvely adding 'ghost-particles' as in the attraction-repulsion experiment did not improve results notably. Interestingly, the power law exponent was still very well recovered.

### 2.3.3 COULOMB-LIKE

Supplementary Figure 8c shows the results of the GNN trained with simulations of particles following Coulomb's law of charge-based attraction-repulsion (see Figure 2c, Supplementary Table 1 and Appendix A.1.1), using the same short-range approximation as previously.

We trained on a series of 2,000 time-steps with 960 particles of three different charges (-1, 1, 2 in arbitrary units). The learnable pairwise interaction function symmetrically depends on the observable relative positions of two particles and both of their latent charges, leading to five distinct interaction profiles. The GNNs trained with this dataset do not yield precise rollout inference owing to error accumulation (see Video 7). However symbolic regression is able to recover the $1/d_{ij}^2$ power laws and scaling scalars. Assuming that the extracted scalars correspond to products $q_i q_j$, it is possible to find the set of $q_i$ values corresponding to these products. Using gradient descent, we recovered the correct $q_i$ values with a precision of $2 \cdot 10^{-2}$. To obtain these results, it was necessary to increase the hidden dimension of the learnable MLP from 128 to 256.

Adding noise during training had limited effect up to $\sigma = 0.3$ (see Supplementary Figure 9c, d), but removing particles from the training data was detrimental and naïvely adding 'ghost-particles' did not improve the results notably (Supplementary Figure 10c, d).

### 2.3.4 BOIDS

Supplementary Figure 11 and Video 8 show the process of training a GNN with the boids simulation (see Figure 2d, Supplementary Table 1 and Appendix A.1.1). We trained on a series of 8,000 time-steps and 1,792 particles with 16, 32, and 64 types, respectively (see Supplementary Figure 12 and Video 9). After training the GNN, we applied hierarchical clustering to the learned latent vectors $a_i$ which separated the particle types with an accuracy of $\sim 1.00$. In Supplementary Figure 13, we show rollout examples for the dynamic behavior of each individual recovered type in isolation. We believe that this 'virtual decomposition' of dynamical systems that can only be observed in mixed configurations will become a powerful tool to understand complex physical, chemical and biological processes. We were not able to recover the multi-variate interaction functions using symbolic regression. However, for the correct symbolic interaction function, we could estimate the latent parameters for each particle type using robust regression (see Supplementary Figure 12). Adding noise during training has a limited effect up to $\sigma = 0.4$ and, anecdotally, even improved parameter estimates. Removing trajectories was detrimental to the ability to learn and reproduce the behavior and parameters (see Supplementary Figure 14).

### 2.3.5 WAVE-PROPAGATION AND DIFFUSION

We simulated wave-propagation and reaction-diffusion processes over a mesh of $10^4$ nodes (see Figure 2e, f, Supplementary Table 1 and Appendix A.1.3). Other than in the simulations with moving

particles, we hard-coded the interaction and aggregation functions to be the discrete Laplacian $\nabla^2$ and learned first (diffusion) and second order updates (wave) with an MLP that has access to the Laplacian, the node's state, and a learnable latent vector $\boldsymbol{a}_i$.

Supplementary Figures 15 and 16 show our results with the wave-propagation model trained on a series of 8,000 time-steps and $10^4$ nodes. We varied the wave-propagation coefficients in discrete patches (see Supplementary Figure 15) and arbitrarily (see Supplementary Figure 16). The GNN correctly recovers the update functions for every nodes, and the linear dependence of these function over the Laplacian of the node states allows to extract correctly all $10^4$ latent propagation-coefficients (slope= 0.96, $R^2 = 0.95$, see Video 11). Rollout inference captures the dynamics of the system qualitatively, but diverges after 3,000 time-steps owing to error accumulation (see Video 12).

Supplementary Figure 17 shows the results of a similar experiment with the "Rock-Paper-Scissors" (RPS) reaction-diffusion simulation (see Video 13). In this model, the nodes are associated with three states $\{u_i, v_i, w_i\}$. The first-time derivatives of these states evolve according to three cyclic equations involving the Laplacian operator and a polynomial function of degree 2 (see Supplementary Table 1). We varied the diffusion coefficient in discrete patches. The GNN correctly identifies four distinct clusters in the latent vectors domain (see Supplementary Figure 17c) that can be mapped over the node positions (see Supplementary Figure 17d). The four recovered types can be analysed separately. For each type, we estimated the diffusion coefficients and the polynomial function coefficients using robust regression. In total, 31 coefficients describing the reaction-diffusion process were accurately retrieved (see Supplementary Figure 17e, f, slope= 1.00, $R^2 = 1.00$).

### 2.3.6 SIGNALING NETWORKS

Supplementary Figure 18 shows our results to recover the rules of a synaptic signaling model with 998 nodes and 17,865 edges with two types of nodes with distinct interaction function (data from Hens et al., 2019; see Figure 2g, Supplementary Table 1, and Video 14). In order for the GNN to successfully infer the signaling rules, we found that we needed a relatively large training dataset. We ran 100 simulations with different initial states over 1,000 time-steps. In addition, it was necessary to predict more than a single time-step to efficiently train the GNN (see Appendix A.1.4). With this training scheme, we recovered the connectivity matrix (slope = 1.0, $R^2 = 1.0$) and were able to automatically infer and parameterize the symbolic interaction functions for both node types using the PySR package (Cranmer, 2023; Lemos et al., 2023).

### 2.4 DISCUSSION

We showed with a diverse set of simulations that message passing GNNs that jointly learn interaction- and update functions and latent node properties are a flexible tool to predict, decompose, and eventually understand complex dynamical systems. With the currently available software libraries (PyTorch Geometric, Fey & Lenssen, 2019), it is straightforward to implement an architecture and loss that encode useful assumptions about the structure of the complex system such as local connectivity rules or the location of learnable and known functions and their inputs. We showed that a well designed GNN can learn a low-dimensional embedding of complex latent properties required to parameterize heterogeneous particle-particle interactions. The learned low-dimensional embeddings can be used to reveal the structure of latent properties underlying the complex dynamics and to infer the corresponding parameters. As demonstrated by Lemos et al. (2023) and in the signaling network experiment, it is possible to use automatic methods to extract symbolic hypotheses that are consistent with the learned dynamics. However, even without an explicit analysis of the underlying functions, it is possible to dissect the dynamical system and to conduct virtual experiments with arbitrary compositions (or decompositions) of particles and interactions. We believe this ability to become particularly useful to infer the local rules governing complex biological systems like the organization of bacterial communities, embryonic development, neural networks, or the social interactions governing animal communities that cannot be observed in isolation.

In preparation for applications on experimental data, we designed simulations that provide some of the required components to model a complex biological process. We demonstrated the ability to reconstruct discrete, continuous, and time-changing heterogeneities. We modeled dynamic interactions between moving agents that interact with each other and with an independent dynamic environment. We were also able to infer the connectivity of a signaling network from functional observations alone.

However, for an application in biology, some key features are still missing and we are planning to develop them in future work:

1. In our simulations, the latent properties are either static or follow an internal program. In biological systems, the interaction between cells and the environment changes their properties over time in well-defined ways.

2. A community of cells interacts with the environment by receiving and releasing signals from and into the environment. Our models currently cover only one direction of communication but will be easy to extend.

3. With the exception of the learned movie experiment (see Figure 4), our models have no memory and do not integrate information over time. There are many ways to implement memory, and it is important to understand that almost all dynamics can be learned by simply memorizing them. We will have to design architectures and training paradigms that avoid this shortcut.

4. In a developing multi-cellular organism, individual cells both divide and die.

5. In our current experiments, we have simulated and learned deterministic functions (with noise). In complex biophysical systems in the real world, it is more likely that interactions are probabilistic and have complex posterior distributions that cannot be learned by regressing to the mean.

## 3 CONCLUSION

We demonstrated that message passing GNNs can learn to replicate the behavior of complex heterogeneous dynamical systems and to uncover the underlying latent properties in an interpretable way that facilitates further analysis. The flexibility in designing GNNs to model and train meaningful interactions in complex systems is impressive and we are looking forward to developing them as an integral toolbox to uncover the rules governing complex dynamics observed in nature.

## 4 ACKNOWLEDGEMENTS

Anonymous acknowledgements, paper under double-blind review.

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

# A APPENDIX

## A.1 GNN IMPLEMENTATION WITH PRIORS

The general form of the GNN update rule described in Section 2.2 and Equation 2 does not consider application specific parameterization of the individual functions. As described in Section 2.2 and Supplementary Table 2, we define the loss over first and second order updates, respectively. For clarity, we unroll here the combination of interaction function, aggregation, and update rules and their respective parameterization, resulting in one equation to calculate the updates.

### A.1.1 PARTICLE SYSTEMS

The gravity-like and the Coloumb-like model use sum as an aggregator $\bigodot$, the attraction-repulsion and boids model use the average. In addition to the latent learnable vector $\boldsymbol{a}_i$, we parameterized the interaction function with the relative distance $d_{ij}$ and relative positions $\boldsymbol{x}_j - \boldsymbol{x}_i$ of the two particles making them blind to absolute position and other non-local information. Hence, the update rules to be learned by the GNNs become

$$\dot{\boldsymbol{x}}_i = \bigodot f(\boldsymbol{a}_i, d_{ij}, \boldsymbol{x}_j - \boldsymbol{x}_i) \quad \text{or}$$

$$\ddot{\boldsymbol{x}}_i = \bigodot f(\boldsymbol{a}_i, d_{ij}, \boldsymbol{x}_j - \boldsymbol{x}_i),$$

respectively. The learnables are the function $f$ and the particle embedding $\boldsymbol{a}_i$. For $f$, we used a five-layer MLP with ReLU activation layers for a total of 50,562 trainable parameters (hidden dimension = 128, input dimension = 5, output dimension = 2). The latent vectors $\boldsymbol{a}_i$ have two dimensions. All particles within a maximum radius are connected. Our particles have no size and we did not model interactions between physical bodies. Interactions governed by inverse power laws have a singularity at $r = 0$ which breaks optimization by gradient descent. We therefore had to limit connectivity to $d_{ij} > 0.02$ for the gravity-like and Coulomb-like models.

In the Coulomb-like system, the interaction functions have both particle embeddings $\boldsymbol{a}_j$ and $\boldsymbol{a}_j$ as input values, and the corresponding update rule is given by

$$\ddot{\boldsymbol{x}}_i = \bigodot f(\boldsymbol{a}_i, \boldsymbol{a}_j, d_{ij}, \boldsymbol{x}_j - \boldsymbol{x}_i).$$

For $f$, we used a five-layer MLP with ReLU activation layers for a total of 190,752 trainable parameters (hidden dimension = 256, input dimension = 7, output dimension = 2).

In the boids model, $f$ has access to the particle velocities, yielding the update rule

$$\ddot{\boldsymbol{x}}_i = \bigodot f(\boldsymbol{a}_i, d_{ij}, \boldsymbol{x}_j - \boldsymbol{x}_i, \dot{\boldsymbol{x}}_i, \dot{\boldsymbol{x}}_j).$$

For $f$, we used a five-layer MLP with ReLU activation layers for a total of 200,450 trainable parameters (hidden dimension = 256, input dimension = 9, output dimension = 2).

### A.1.2 PARTICLE SYSTEMS AFFECTED BY A HIDDEN FIELD

We added a latent external process as a hidden field of $10^4$ randomly distributed stationary particles that interact with the moving particles using the same interaction rules and distance based neighborhood. The stationary particles show either a constant image or a time-variant movie that define a latent coefficient $b_j$ that modulates the interaction. For simplicity, we can write the interaction between all particles with this coefficient $b_j$, knowing that for interactions between moving particles $b_j = 1$.

$$\dot{\boldsymbol{x}}_i = \bigodot b_j f(\boldsymbol{a}_i, d_{ij}, \boldsymbol{x}_j - \boldsymbol{x}_i).$$

We learn $b_j$ using an MLP with the position $\boldsymbol{x}_j$ and the time-index $t$ (for time-variant processes) as inputs:

$$\dot{\boldsymbol{x}}_i = \bigodot b(\boldsymbol{x}_j, t) f(\boldsymbol{a}_i, d_{ij}, \boldsymbol{x}_j - \boldsymbol{x}_i).$$

We used an MLP with periodic activation functions (Sitzmann et al., 2020) and 5 hidden layers of dimension 128. The total number of trainable parameters for this MLP is 83,201 (hidden dimension = 128, input dimension = 3, output dimension = 1).

### A.1.3 SIMULATION OF WAVE-PROPAGATION AND REACTION-DIFFUSION

We used the Eulerian representation for the simulations of wave-propagation and reaction-diffusion. The particle positions are fixed and a vector $\boldsymbol{u}_i$ associated with each node evolves over time. All particles are connected to their neighbors using Delaunay triangulation. We fixed the interaction and aggregation functions to be the discrete mesh Laplacian over this neighborhood. For wave-propagation, we learn the second time-derivative of $\boldsymbol{u}_i$

$$\ddot{\boldsymbol{u}}_i = \Phi\Big(a_i, \nabla^2 \boldsymbol{u}_i\Big).$$

For $\Phi$, we used a five-layer MLP with ReLU activation layers for a total of 897 trainable parameters (hidden dimension = 16, input size = 3, output size = 1).

For the reaction-diffusion simulation, we learn the first time-derivative of $\boldsymbol{u}_i$

$$\dot{\boldsymbol{u}}_i = \Phi\Big(a_i, \boldsymbol{u}_i, \nabla^2 \boldsymbol{u}_i\Big).$$

$\nabla^2 \boldsymbol{u}_i$ can not be calculated on the edges, so we discarded the borders during training (1164 nodes out of $10^5$). For $\Phi$, we used a five-layer MLP with ReLU activation layers for a total of 4,422 trainable parameters (hidden dimension = 32, input size = 5, output size = 3).

### A.1.4 SIGNALING

The signaling network is described by a set of nodes without position information connected according to a symmetric connectivity matrix $\boldsymbol{A}$. We learn

$$\dot{u}_i = \Phi(\boldsymbol{a_i}, u_i) + \sum_{j \in V_i} \boldsymbol{A}_{ij} f(u_j)$$

as described by Hens et al. (2019); Aguirre & Letellier (2009); Gao et al. (2016); Karlebach & Shamir (2008); Stern et al. (2014). The learnables are the functions $\Phi$ and $f$, the node embedding $\boldsymbol{a}_i$, and the connectivity matrix $\boldsymbol{A}$ ($960 \times 960$). For $\Phi$, we used a three-layer MLP with ReLU activation layers for a total of 4,481 trainable parameters (hidden dimension = 64, input size = 3, output size = 1). For $f$, we used a three-layer MLP with ReLU activation layers for a total of 4,353 trainable parameters (hidden dimension = 64, input size = 1, output size = 1). Symmetry of $\boldsymbol{A}$ was enforced during training resulting in 17,865 learnable parameters. Since no data augmentation is possible, we generated 100 randomly initialized training-series. Training was unstable for unconstrained next time-step prediction, but stable when training to predict at least two consecutive time-steps. The loss for two consecutive time-steps is

$$L_{\dot{u},t} = \sum_{i=1}^{n} (\|\widehat{\dot{u}}_{i,t+1} - \dot{u}_{i,t+1}\|^2 + \|\widehat{\dot{u}}_{i,t+2} - \dot{u}_{i,t+2}\|^2),$$

with $\widehat{\dot{u}}_{i,t+2}$ calculated after updating

$$u_{i,t+1} \leftarrow u_{i,t+1} + \Delta t\, \widehat{\dot{u}}_{i,t+1}.$$

### A.2 CLUSTERING OF GNN'S LATENT VECTORS AND LEARNED INTERACTION FUNCTIONS

The latent vectors or the learned interaction functions are clustered using the SciPy library (Virtanen et al., 2020). We used the Euclidean distance metric to calculate distances between points and performed hierarchical clustering using the single linkage algorithm, and formed flat clusters using the distance method with a cut-off threshold of 0.01. To cluster the interaction functions, their profiles are first projected to two dimensions using UMAP dimension reduction. The UMAP projections are next clustered to obtain the different classes of interaction functions.

Supplementary Table 1: Description of the simulations.

| Description | Observables | Connectivity $V_i$ | Interaction | Update |
|---|---|---|---|---|
| Attraction-repulsion | $\boldsymbol{x}_i \in [0,1)^2$ | $d_{ij} \in (0.002, 0.075)$ periodic | $f_{ij} = a_i g(d_{ij}, b_i) - c_i g(d_{ij}, d_i)$ $g(x,y) = \exp(-x^{2y}/2\sigma^2)$ $a_i, b_i, c_i, d_i \in [1,2]$ $\sigma = 0.005$ | $\dot{\boldsymbol{x}}_i \leftarrow \frac{1}{|V_i|} \sum_{j \in V_i} f_{ij}$ $\boldsymbol{x}_i \leftarrow \boldsymbol{x}_i + \Delta t \dot{\boldsymbol{x}}_i$ |
| Gravity-like | $\boldsymbol{x}_i \in \mathbb{R}^2$ | $d_{ij} \in (0.001, 0.3)$ non-periodic | $f_{ij} = m_j(\boldsymbol{x}_j - \boldsymbol{x}_i)/d_{ij}^3$ $m_j \in (0,5]$ | $\ddot{\boldsymbol{x}}_i \leftarrow \sum_{j \in V_i} f_{ij}$ $\dot{\boldsymbol{x}}_i \leftarrow \dot{\boldsymbol{x}}_i + \Delta t \ddot{\boldsymbol{x}}_i$ $\boldsymbol{x}_i \leftarrow \boldsymbol{x}_i + \Delta t \dot{\boldsymbol{x}}_i$ |
| Coulomb-like | $\boldsymbol{x}_i \in [0,1)^2$ | $d_{ij} \in (0.001, 0.3)$ periodic | $f_{ij} = -q_i q_j(\boldsymbol{x}_j - \boldsymbol{x}_i)/d_{ij}^3$ $q_i, q_j \in [-2, 2]$ | $\ddot{\boldsymbol{x}}_i \leftarrow \sum_{j \in V_i} f_{ij}$ $\dot{\boldsymbol{x}}_i \leftarrow \dot{\boldsymbol{x}}_i + \Delta t \ddot{\boldsymbol{x}}_i$ $\boldsymbol{x}_i \leftarrow \boldsymbol{x}_i + \Delta t \dot{\boldsymbol{x}}_i$ |
| Boids | $\boldsymbol{x}_i \in [0,1)^2$ | $d_{ij} \in (0.001, 0.04)$ periodic | $f_{ij} = \boldsymbol{a}_{ij} + \boldsymbol{c}_{ij} + \boldsymbol{s}_{ij}$ $\boldsymbol{c}_{ij} = c_i(\boldsymbol{x}_j - \boldsymbol{x}_i)$ $\boldsymbol{a}_{ij} = a_i(\dot{\boldsymbol{x}}_j - \dot{\boldsymbol{x}}_i)$ $\boldsymbol{s}_{ij} = -s_i(\boldsymbol{x}_j - \boldsymbol{x}_i)/d_{ij}^2$ $a_i, c_i, s_i \in \mathbb{R}$ | $\ddot{\boldsymbol{x}}_i \leftarrow \frac{1}{|V_i|} \sum_{j \in V_i} f_{ij}$ $\dot{\boldsymbol{x}}_i \leftarrow \dot{\boldsymbol{x}}_i + \Delta t \ddot{\boldsymbol{x}}_i$ $\boldsymbol{x}_i \leftarrow \boldsymbol{x}_i + \Delta t \dot{\boldsymbol{x}}_i$ |
| Wave-propagation | $u_i \in \mathbb{R}$ | Delaunay non-periodic | $\nabla^2 u_i$ | $\ddot{u}_i \leftarrow a_i \nabla^2 u_i, \quad a_i \in [0,1]$ $\dot{u}_i \leftarrow \dot{u}_i + \Delta t \ddot{u}_i$ $u_i \leftarrow u_i + \Delta t \dot{u}_i$ |
| Reaction-diffusion Rock-Paper-Scissors (RPS) model | $u_i, v_i, w_i \in (0,1]^3$ | Delaunay non-periodic | $\nabla^2 u_i, \nabla^2 v_i, \nabla^2 w_i$ | $\dot{u}_i \leftarrow a_i \nabla^2 u_i + u_i(1 - p_i - \beta v_i)$ $\dot{v}_i \leftarrow a_i \nabla^2 v_i + v_i(1 - p_i - \beta w_i)$ $\dot{w}_i \leftarrow a_i \nabla^2 w_i + w_i(1 - p_i - \beta u_i)$ $u_i \leftarrow u_i + \Delta t \dot{u}_i$ $v_i \leftarrow v_i + \Delta t \dot{v}_i$ $w_i \leftarrow w_i + \Delta t \dot{w}_i$ $p_i = u_i + v_i + w_i$ $\beta \in [0,1], a_i \in [0,1]$ |
| Signaling | $u_i \in \mathbb{R}$ | Connectivity matrix $\boldsymbol{A} \in \mathbb{R}^{n \times n}$ | $f_{ij} = \boldsymbol{A}_{ij} \tanh(u_j)$ | $\dot{u}_i \leftarrow -b_i u_i + c_i \cdot \tanh(u_i) + \sum_{j \in V_i} f_{ij}$ |

$\boldsymbol{x}_i$ denotes the two-dimensional coordinate vector associated with particle $i$. The distance between particles $i$ and $j$ is given by $d_{ij} = \|\boldsymbol{x}_j - \boldsymbol{x}_i\|$. In the arbitrary attraction-repulsion model, the interaction is parameterized by the coefficients $a_i, b_i, c_i, d_i$, that are different for each particle or particle type. In the gravity-like model, each particle has a mass $m_i$. In the Coulomb-like model, each particle type has a charge $q_i$. The boids interaction function consists of three terms, cohesion $\boldsymbol{c}_{ij}$, alignment $\boldsymbol{a}_{ij}$, and separation $\boldsymbol{s}_{ij}$, parameterized by $c_i, a_i$, and $s_i$, respectively. In the wave-propagation, Rock-Paper-Scissors (RPS), and signaling models, particles are stationary and their neighborhood remains constant. In the wave-propagation model, a scalar property $u_i$ of each particle evolves over time. Information from connected particles is integrated via discrete Laplacian $\nabla^2$. Each particle has a unique latent coefficient $\alpha_i$ that modulates the wave-propagation. The RPS model is a reaction-diffusion model over three-dimensional dynamic particle properties. Similar to the speed of wave-propagation model, diffusion is realized via discrete Laplacian $\nabla^2$, and the complex reaction function for each particle is modulated by a latent diffusion factor $\alpha_i$. The signaling model is an activation model for dynamics between brain regions as discussed in (Stern et al., 2014). The signal per particle is an activation variable $u_i$ that propagates through the network defined by a symmetric connectivity matrix $\boldsymbol{A}$. The update function at each particle is modulated by two latent coefficients $b_i$ and $c_i$.

Supplementary Table 2: Architecture and parameterization of the GNNs used to predict and decompose the simulations detailed in Supplementary Table 1.

| Description | Interaction | Aggregation | Update |
|---|---|---|---|
| Attraction-repulsion | $f_{ij} = \mathrm{MLP}(\boldsymbol{a}_i, d_{ij}, \boldsymbol{x}_j - \boldsymbol{x}_i)$ | $\odot_i = \frac{1}{|V_i|} \sum\limits_{j \in V_i} f_{ij}$ | $\widehat{\dot{\boldsymbol{x}}}_i = \odot_i$ |
| Gravity-like | $f_{ij} = \mathrm{MLP}(\boldsymbol{a}_i, d_{ij}, \boldsymbol{x}_j - \boldsymbol{x}_i, \dot{\boldsymbol{x}}_i, \dot{\boldsymbol{x}}_j)$ | $\odot_i = \sum\limits_{j \in V_i} f_{ij}$ | $\widehat{\ddot{\boldsymbol{x}}}_i = \odot_i$ |
| Coulomb-like | $f_{ij} = \mathrm{MLP}(\boldsymbol{a}_i, \boldsymbol{a}_j, d_{ij}, \boldsymbol{x}_j - \boldsymbol{x}_i)$ | $\odot_i = \sum\limits_{j \in V_i} f_{ij}$ | $\widehat{\ddot{\boldsymbol{x}}}_i = \odot_i$ |
| Boids | $f_{ij} = \mathrm{MLP}(\boldsymbol{a}_i, d_{ij}, \boldsymbol{x}_j - \boldsymbol{x}_i, \dot{\boldsymbol{x}}_i, \dot{\boldsymbol{x}}_j)$ | $\odot_i = \frac{1}{|V_i|} \sum\limits_{j \in V_i} f_{ij}$ | $\widehat{\dot{\boldsymbol{x}}}_i = \odot_i$ |
| Wave-propagation | | $\odot_i = \nabla^2 u_i$ | $\widehat{\ddot{u}}_i = \mathrm{MLP}(\boldsymbol{a}_i, u_i, \odot_i)$ |
| Reaction-diffusion | | $\odot_i = \nabla^2 \boldsymbol{u}_i$ | $\widehat{\dot{\boldsymbol{u}}}_i = \mathrm{MLP}(\boldsymbol{a}_i, \boldsymbol{u}_i, \odot_i)$ |
| Signaling | $f_{ij} = \boldsymbol{A}_{ij} \cdot \mathrm{MLP}(u_j)$ | $\odot_i = \sum\limits_{j \in V_i} f_{ij}$ | $\widehat{\dot{u}}_i = \mathrm{MLP}(\boldsymbol{a}_i, u_i) + \odot_i$ |

Each GNN is characterized by its input parameters and its learnable (red) and fixed parameters and functions (black). For update functions, we show the predicted first or second order property that is used to calculate the training loss and to update particle states using explicit or semi-implicit Euler integration (see Equation 1). The models for moving particles have a learnable pairwise interaction function with access to a subset of the particle properties and a learnable embedding for either both or one of the particles. The aggregation function is a trivial sum or average over the pairwise interaction functions. For the two propagation models, the pairwise interaction and aggregation functions combined are the discrete Laplacian $\nabla^2$. Here, the update function and an embedding for the latent node properties are learnable. The model for the signaling network simulation has a learnable pairwise interaction function and a learnable update function with access to the learnable connectivity matrix and a learnable embedding for the coefficients modulating the update.

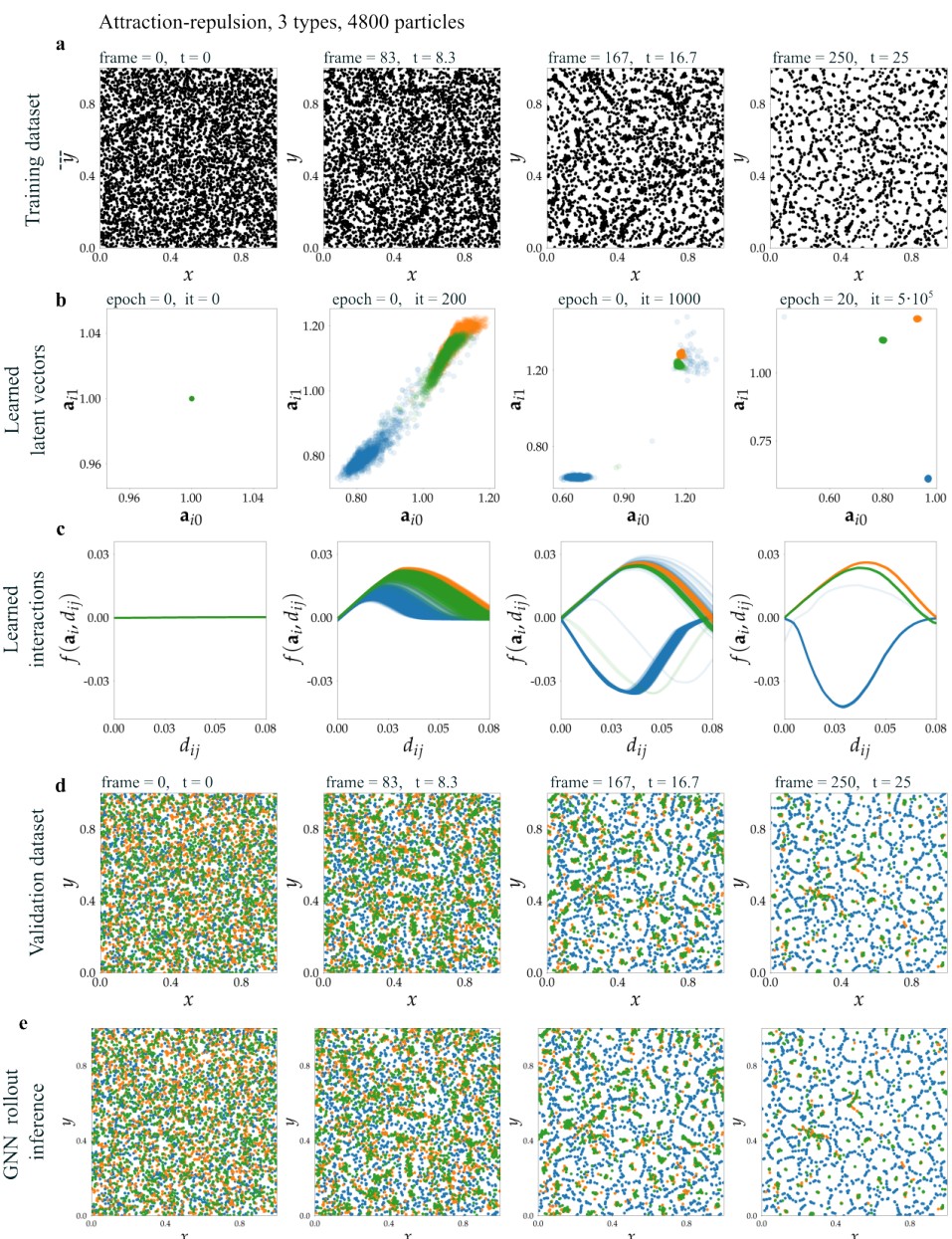

Supplementary Figure 1: GNN trained on an attraction-repulsion simulation (4,800 particles, 3 particle types, 250 time-steps). Colors indicate the true particle type. (**a**) The training dataset is shown in black and white to emphasize that particle types are not known during training. (**b**) Learned latent vectors $a_i$ of all particles as a function of epoch and iteration number. Colors indicate the true particle type. (**c**) Projection of the learned interaction functions $f$ as speed over distance. (**d**) Validation dataset with initial conditions different from training dataset. (**e**) Rollout inference of the fully trained GNN. Colors indicate the learned classes found in (**b**).

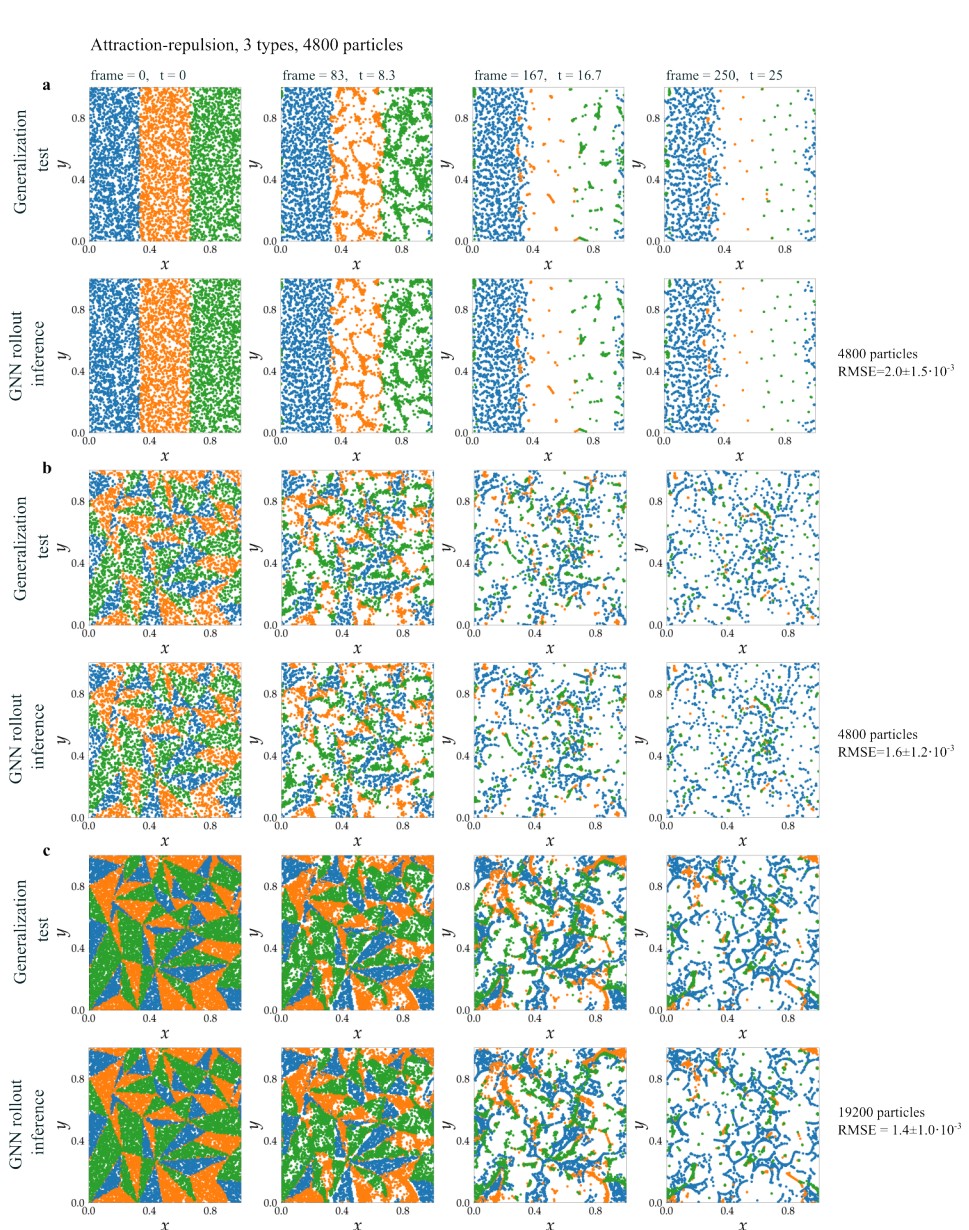

Supplementary Figure 2: Generalization tests successfully run on GNNs trained on attraction-repulsion simulations (4,800 and 19,200 particles, 3 particle types, 250 time-steps). Colors indicate the true particle type. (**a**) Particles are sorted by type into three stripes. Interestingly this test allows to visualize the differences in interactions. (**b**) Particles are sorted by particle type into a triangle pattern. (**c**) Same as (**b**) and the number of particles is increased from 4,800 to 19,200. The additional particles embedding values are sampled from the learned embedding domain. RMSE measured between true positions and GNNs inferences are given, ($\boldsymbol{x}_i \in [0, 1]^2$, 250 time-steps).

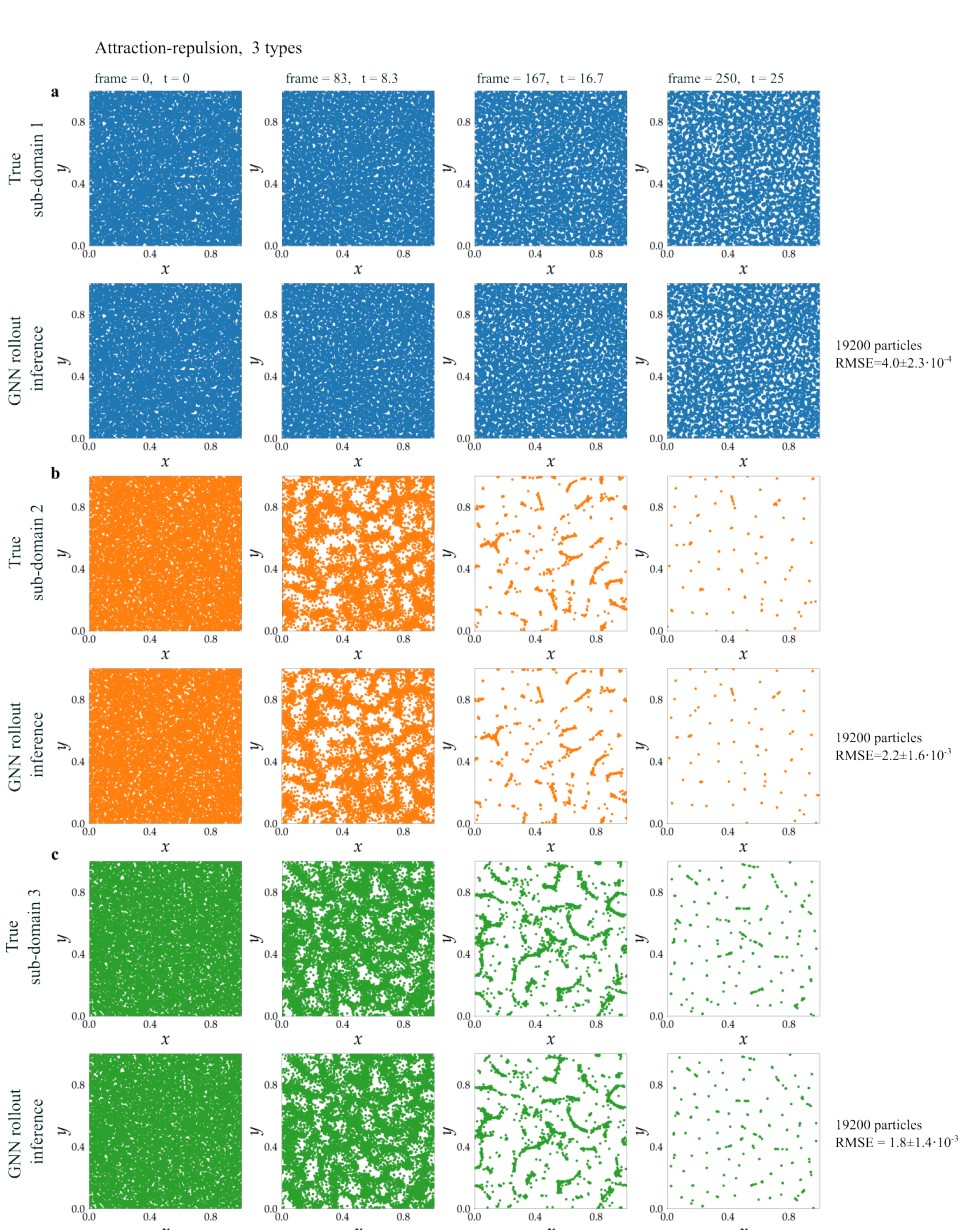

Supplementary Figure 3: Decomposition of the GNN trained on an attraction-repulsion simulation (4,800 particles, 3 particle types, 250 time-steps). Colors indicate the true particle type. The GNN learns to correctly model the three different particle types. The heterogeneous dynamics can be decomposed into 'purified' samples governed by one unique interaction law. (**a**), (**b**), (**c**) show the results for one of three particle types each. True simulated positions are shown in the top row and the rollout inference generated by the GNN are shown in the bottom row. RMSE between true and inferred rollout are shown in the right column ($\boldsymbol{x}_i \in [0, 1]^2$, 19,200 particles, 250 time-steps).

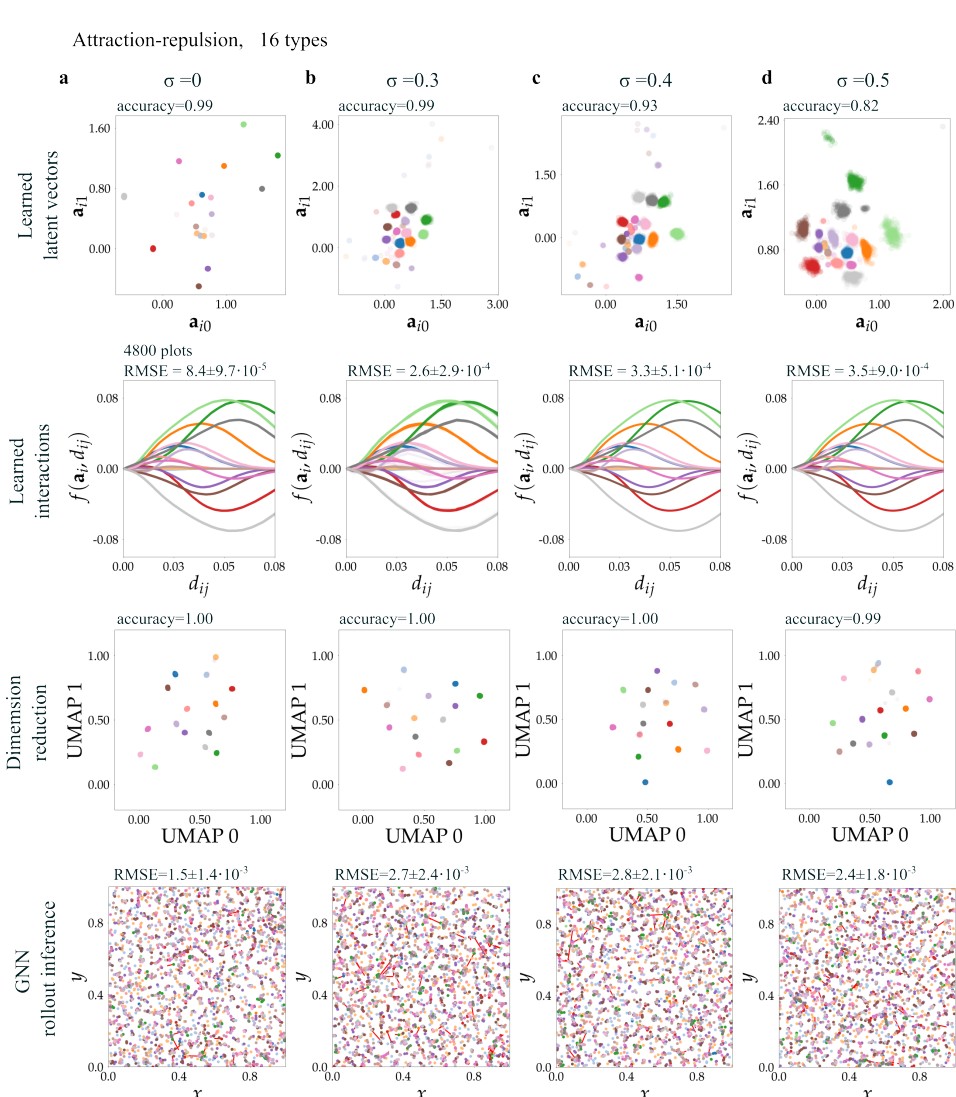

Supplementary Figure 4: Robustness to noise. The GNN was trained with the data of an attraction-repulsion simulation (4,800 particles, 16 particle types, 500 time-steps). Colors indicate the true particle type. To corrupt the training with noise, we used a modified loss $L_{\dot{x}} = \sum_i^n \|\widehat{\dot{x}}_i - \dot{x}_i(1+\varepsilon)\|^2$ where $\varepsilon$ is a random vector drawn from a Gaussian distribution $\varepsilon \sim \mathcal{N}(0, \sigma^2)$. (**a**) to(**d**) show the results for increasing noise $\sigma \in [0, 0.5]$. Row 1 shows the learned particle embedding. Row 2 shows the projection of the learned interaction functions $f$ as speed over distance for each particle. Row 3 shows the UMAP projections of these profiles. Hierarchical clustering of the UMAP projections allows to classify the moving particles with an accuracy of $\sim 1.00$. Row 4 shows the learned particle positions after rollout over 500 time-steps. Colors indicate the learned classes. Deviation from ground truth is shown by red segments.

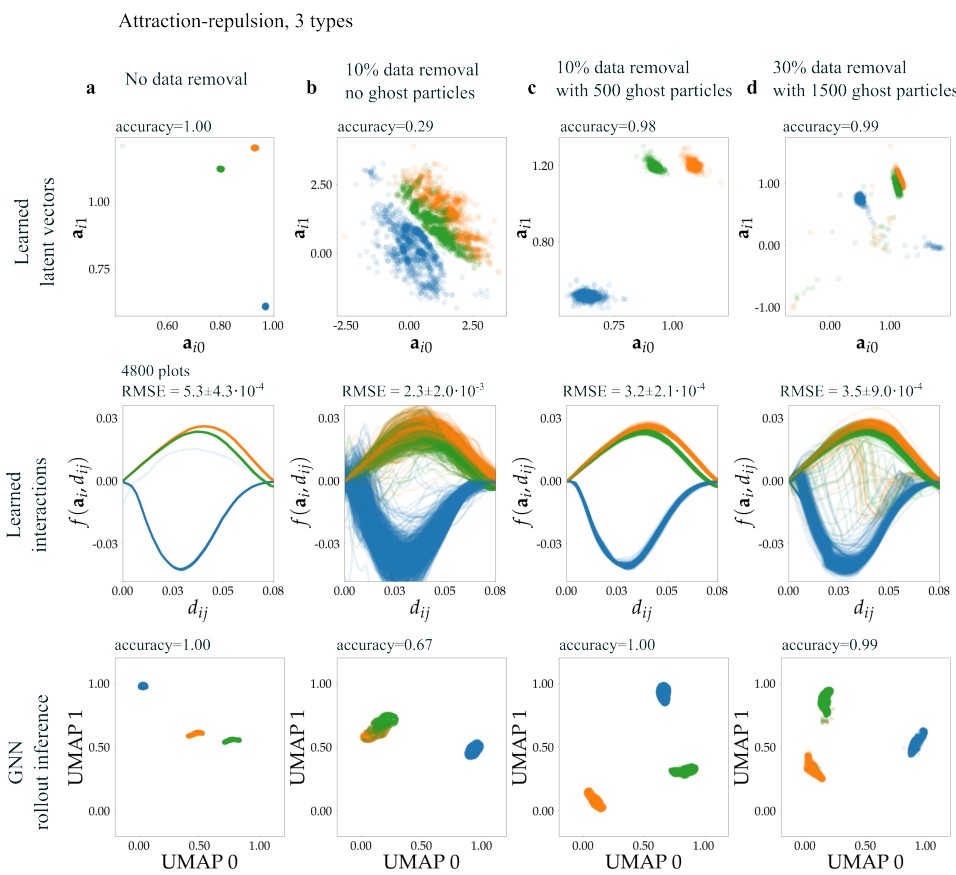

Supplementary Figure 5: Robustness to data removal. Tests are performed with GNNs trained with data of attraction-repulsion simulations (4,800 particles, 3 particle types, 500 time-steps). Four experiments are shown with varying amounts of data removed, with and without the addition of ghost particles during training. Row 1 shows the learned latent vectors $\boldsymbol{a}_i$ of all particles. Row 2 shows the projection of the learned interaction functions $f$ as speed over distance for each particle. Row 3 shows the UMAP projections of these profiles. Hierarchical clustering of the UMAP projections is used to classify particles. Colors indicate the true particle type.

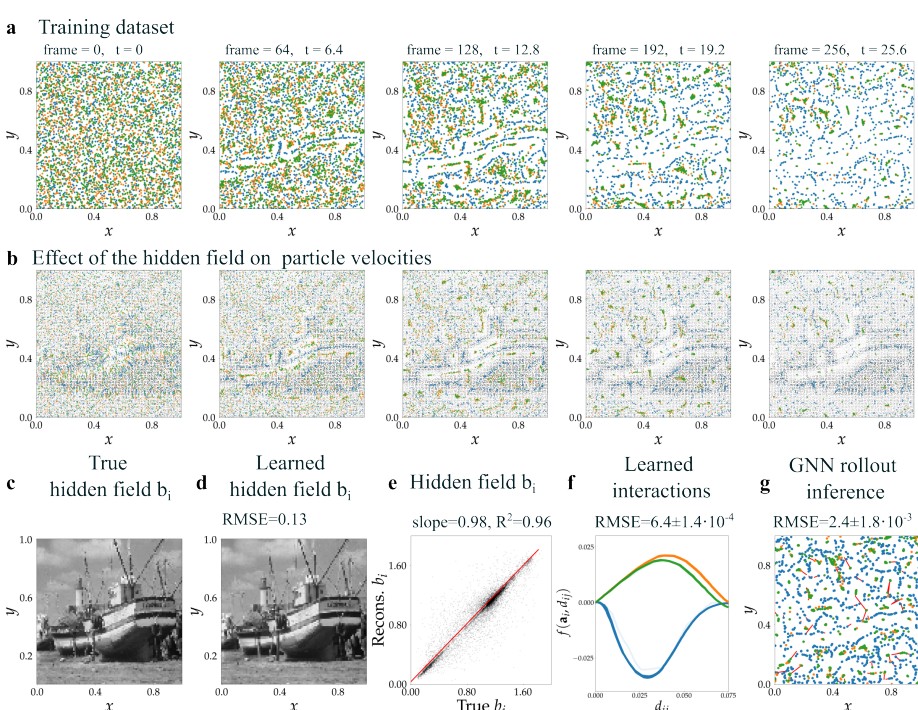

Supplementary Figure 6: Experiment with the attraction-repulsion model whose particles interact with a hidden field. (**a**) Snapshots of the training series with 4,800 particles, 3 particle types (colors) and 256 time-steps. (**b**) Small colored arrows depict the velocities of the moving particles induced by the hidden field (grey dots). (**c**) The external field is simulated by $10^4$ stationary particles with states $b_i$. (**d**) Learned hidden field $b_i$. (**e**) Comparison between learned and true field $b_i$ ($10^4$ points). (**f**) Projection of the learned interaction function $f$ as speed over distance. (**g**) Rollout inference of the trained GNN after 256 time-steps. Colors indicate the learned particle type. Deviation from ground truth is shown by red segments.

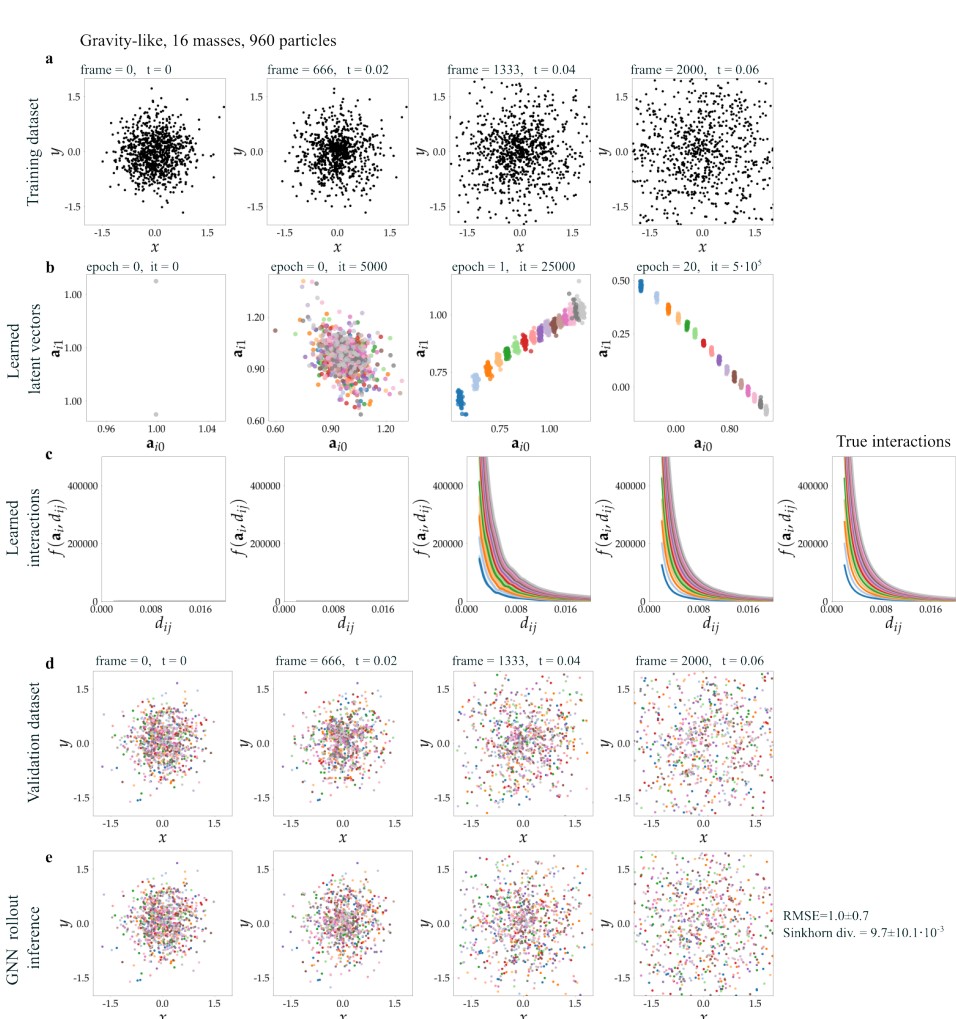

Supplementary Figure 7: GNNs trained on gravity-like simulations (960 particles, 16 masses, 2,000 time-steps). (**a**) The training dataset is shown in black and white to emphasize that particle masses are not known during training. (**b**) The learned latent vectors $\boldsymbol{a}_i$ of all particles. Colors indicate the true masses using an arbitrary color scale. (**c**) the projection of the learned interaction functions $f$ as acceleration over distance for each particle. (**d**) Validation dataset with initial conditions different from training dataset. (**e**) Rollout inference of the trained GNN. Colors indicate the learned classes found in (**b**). The Sinkhorn divergence measures the difference in spatial distributions between ground truth and GNN inference (Feydy et al., 2018).

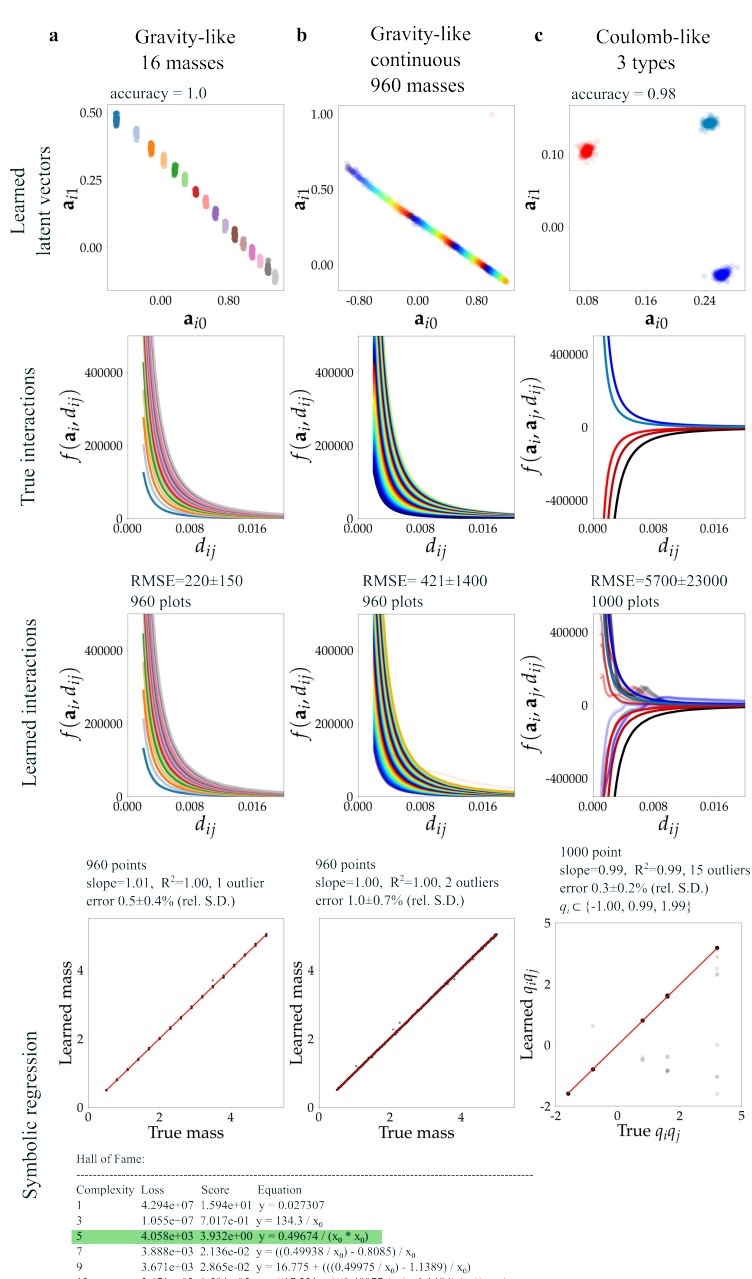

Supplementary Figure 8: GNNs trained on particle systems governed by second-order time derivative equations. (**a**) Gravity-like system with 16 different masses (colors) distributed over 960 particles. (**b**) Gravity-like system with 960 different masses (colors) uniformly distributed over equal number of particles. (**c**) Coulomb-like system with 3 different charges (colors) distributed over 960 particles. Row 1 shows the learned latent vectors $\boldsymbol{a}_i$ of all particles. Row 2 shows the projection of the true interaction functions $f$ as acceleration over distance. Row 3 shows the projection of the learned interaction functions $f$ as acceleration over distance. Row 4 shows the result of symbolic regression (PySR package) applied to the learned interaction functions. As an example, the symbolic regression results are given for the retrieval of the interaction function $0.5/dij^2$. Best result is highlighted in green. The power laws are well recovered with symbolic regression and the extracted scalars are similar to the true mass $m_i$ or products $q_iq_j$. Linear fit and relative errors are calculated after removing outliers. For the Coulomb-like system, we assume that the extracted scalars correspond to products $q_iq_j$. It is then possible to find the set of $q_i$ values corresponding to the extracted products.

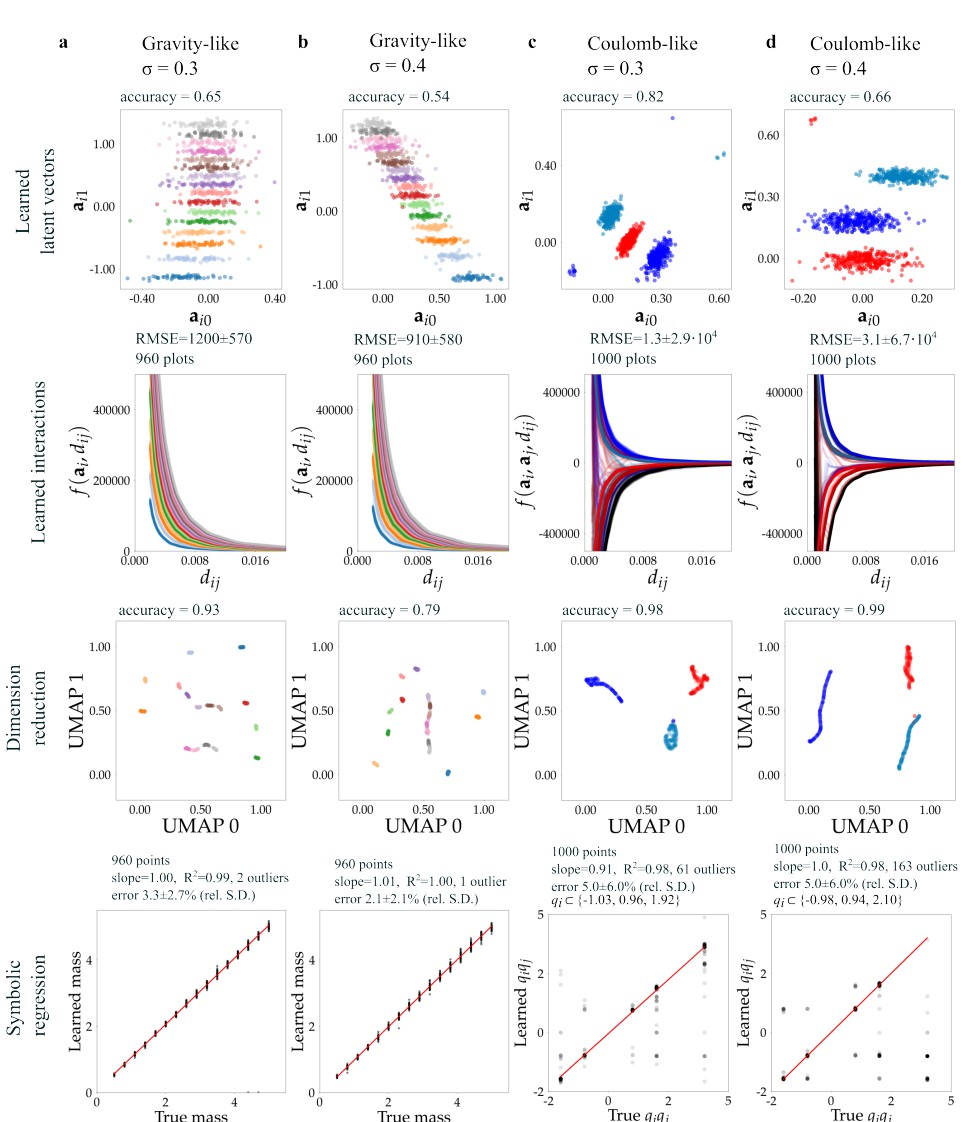

Supplementary Figure 9: Robustness to noise. Tests are performed with the GNN trained on gravity-like and Coulomb-like simulations. To corrupt the training with noise, we used a modified loss $L_{\ddot{\boldsymbol{x}}} = \sum_i^n \|\hat{\ddot{\boldsymbol{x}}}_i - \ddot{\boldsymbol{x}}_i(1 + \boldsymbol{\varepsilon})\|^2$ where $\boldsymbol{\varepsilon}$ is a random vector drawn from a Gaussian distribution $\boldsymbol{\varepsilon} \sim \mathcal{N}(0, \sigma^2)$. Results are shown for $\sigma$ of 0.3 and 0.4. Row 1 shows the learned latent vectors $\boldsymbol{a}_i$ of all particles. Row 2 shows the projection of the learned interaction functions $f$ as acceleration over distance for each particle. Row 3 shows the UMAP dimension reduction of these profiles. Hierarchical clustering of the UMAP projections is used to classify particles. Row 4 shows the result of symbolic regression (PySR package) applied to the learned interaction functions. The power laws are well recovered and the extracted scalars are similar to the true mass $m_i$ or products $q_i q_j$. Linear fit and relative errors are calculated after removing outliers. For the Coulomb-like system, we assume that the extracted scalars correspond to products $q_i q_j$. It is then possible to find the set of $q_i$ values corresponding to the extracted products.

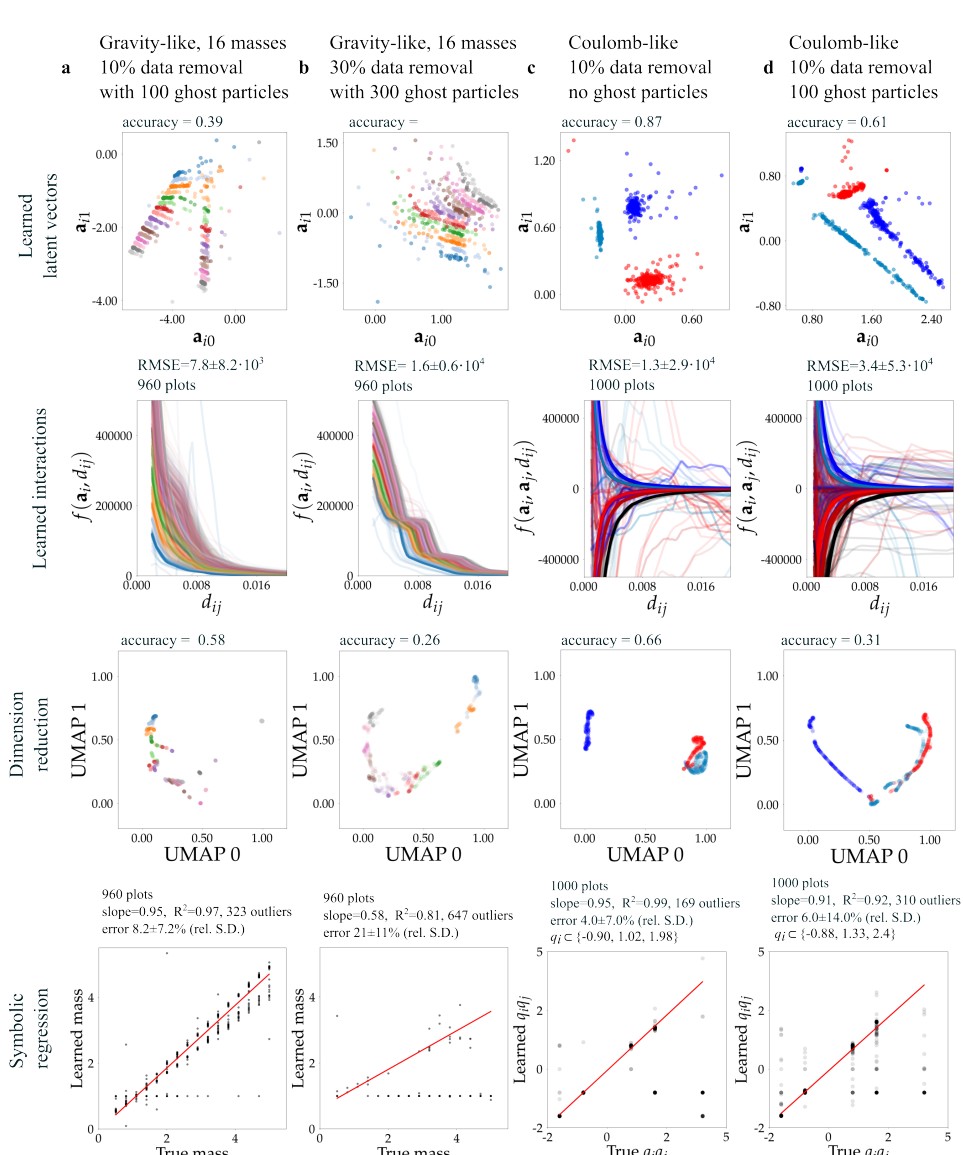

Supplementary Figure 10: Robustness to data removal. Tests are performed with the GNN trained on gravity-like and Coulomb-like simulations. Results are shown for varying amounts of data removed, with and without the addition of ghost particles during training. Row 1 shows the learned latent vectors $\boldsymbol{a}_i$ of all particles. Row 2 shows the projection of the learned interaction functions $f$ as acceleration over distance for each particle. Row 3 shows the UMAP dimension reduction of these profiles. Hierarchical clustering of the UMAP projections is used to classify particles. Row 4 shows the result of symbolic regression (PySR package) applied to the learned interaction functions. When power laws are retrieved, the extracted scalars are similar to the true mass $m_i$ or products $q_i q_j$. Linear fit and relative errors are calculated after removing outliers. For the Coulomb-like system, we assume that the extracted scalars correspond to products $q_i q_j$. It is then possible to find the set of $q_i$ values which can be mapped to the set of extracted scalars.

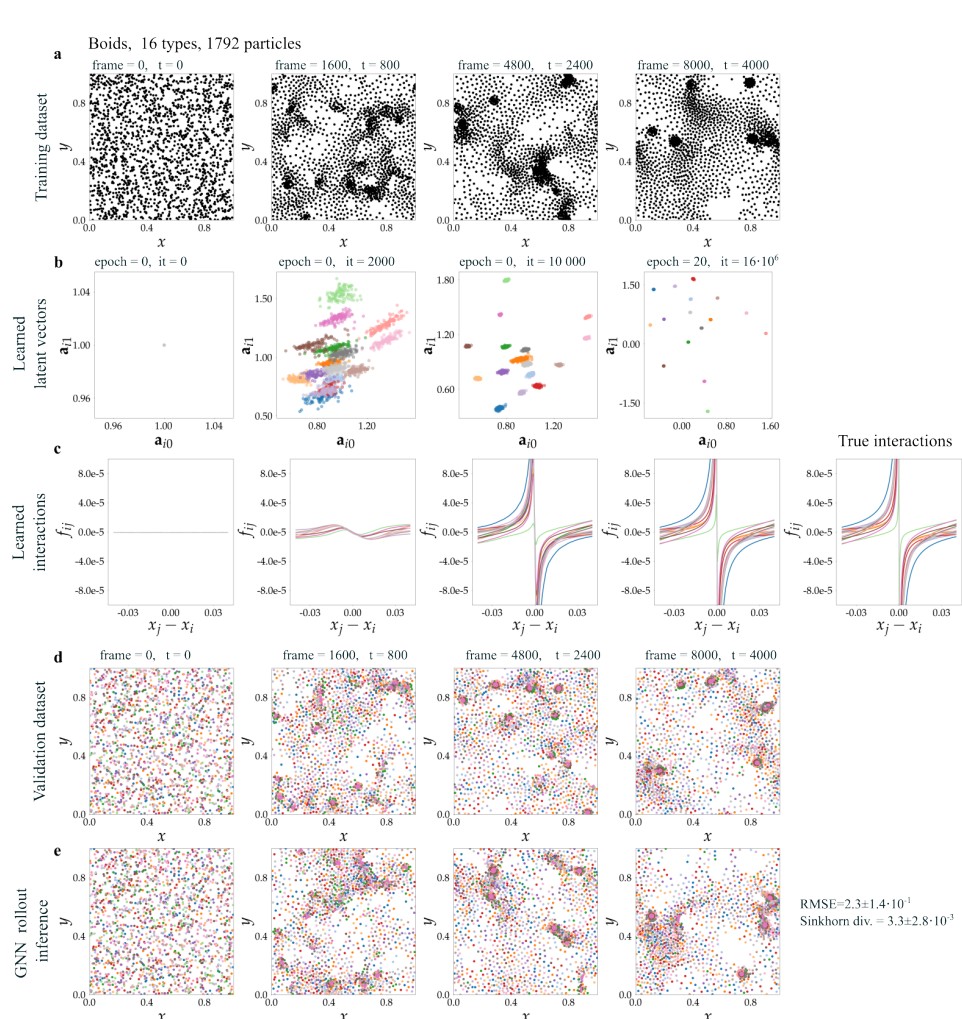

Supplementary Figure 11: GNN trained on a boid simulation (1,792 particles, 16 particle types, 8,000 time-steps). (**a**) The training dataset is shown in black and white to emphasize that particle types are not known during training. (**b**) The learned latent vectors $\boldsymbol{a}_i$ of all particles for different epochs and iterations. Colors indicate the true particle type. (**c**) A projection of the learned interaction functions $f$ as acceleration over distance for each particle. The projections are calculated with the velocity inputs set to 0. **d** Validation dataset with initial conditions different from training dataset. (**e**) Rollout inference of the trained GNN. Colors indicate the learned classes found in (**b**). The Sinkhorn divergence measures the difference in position distributions between ground truth and GNN inference (Feydy et al., 2018).

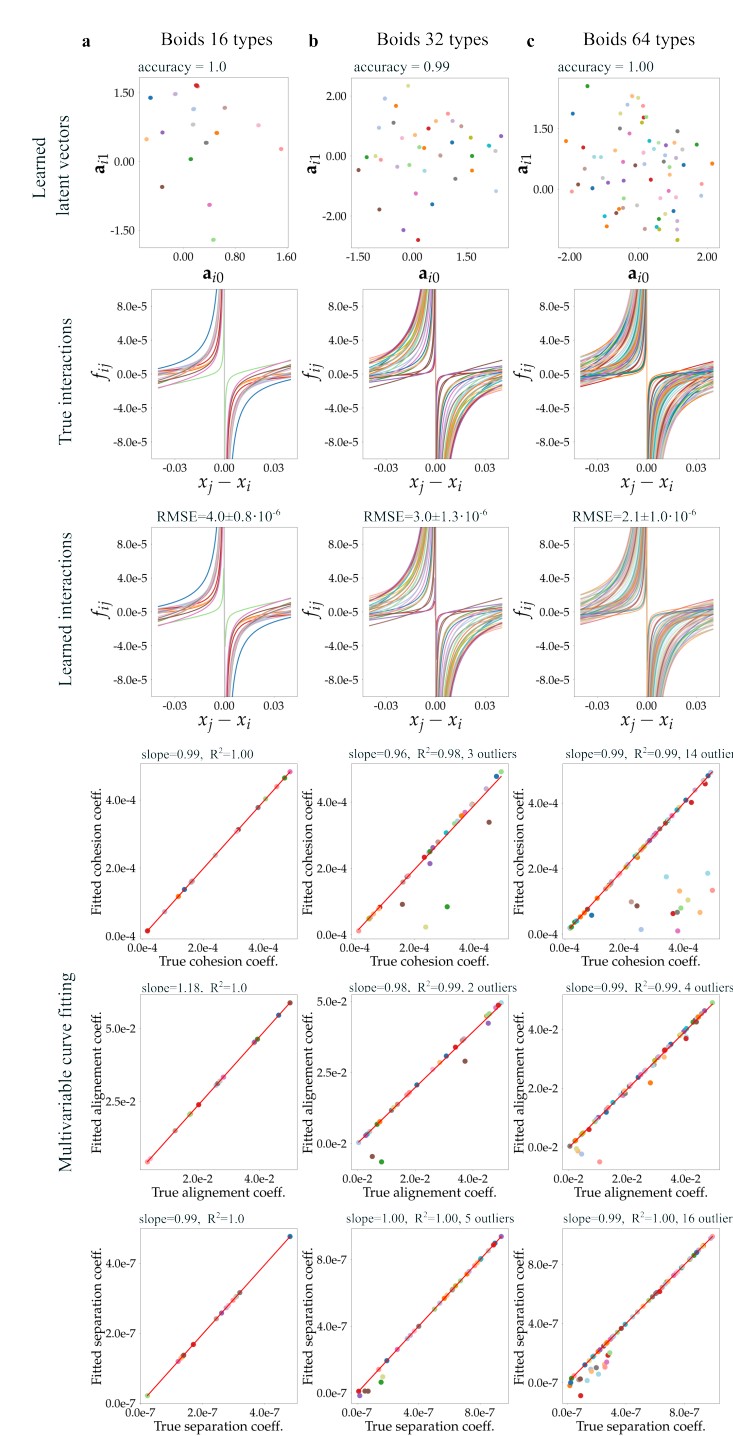

Supplementary Figure 12: GNNs trained on boids simulations (1,792 particles, 16, 32 and 64 particle types, 8,000 time-steps). Row 1 shows the learned latent vectors $\boldsymbol{a}_i$ of all particles. Row 2 shows a projection of the true interaction functions $f$ as acceleration over distance. Row 3 shows the same projection of the learned interaction functions $f$. Rows 4 to 6 show the results of supervised curve fitting of the interaction functions $f$. The correct function $f(\boldsymbol{x_i}, \boldsymbol{x_j}, \dot{\boldsymbol{x_i}}, \dot{\boldsymbol{x_j}}) = a(\boldsymbol{x_j} - \boldsymbol{x_i}) + b(\dot{\boldsymbol{x_j}} - \dot{\boldsymbol{x_i}}) + c(\boldsymbol{x_j} - \boldsymbol{x_i})/d_{ij}^2$ is used to fit the scalars $a$, $b$, $c$, defining cohesion, alignment, and separation (see Supplementary Table 1). Except for a few outliers (rel. error$> 25\%$), all parameters of the simulated boids motions are well recovered.

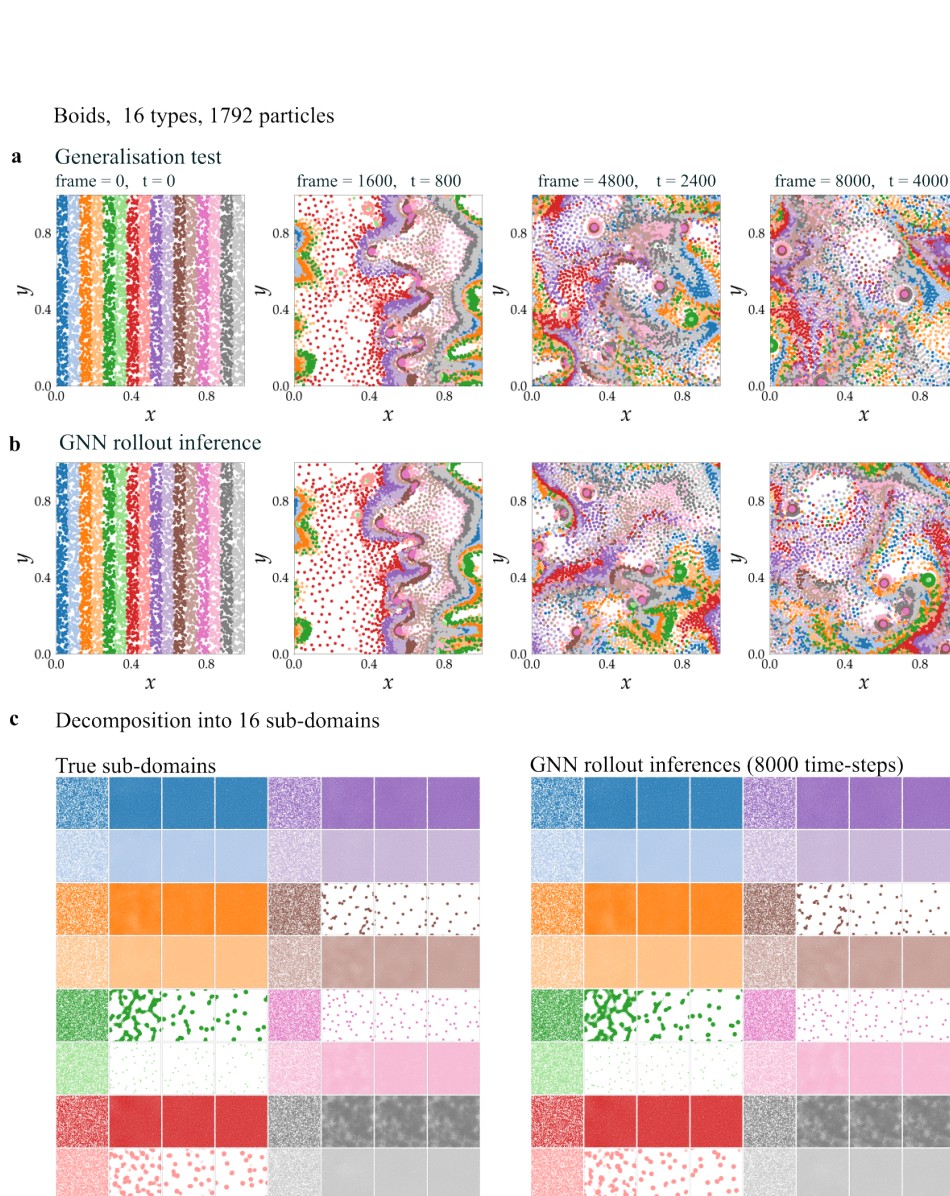

Supplementary Figure 13: (**a**), Generalization and decomposition tests of the GNN trained with the boids simulation (4,800 particles, 16 particle types, 8,000 time-steps). As a generalization test, the number of particle was multiplied by a factor of 4 (from 1,792 to 7,168) and the initial positions were split into 16 stripes to separate particle types. (**a**) shows the ground truth and (**b**) shows the GNN rollout inference. The latter matched ground truth up to 2,000 iterations and remains qualitatively similar later. (**c**) The GNN correctly learned to model the 16 different particle types and their interactions. The heterogeneous dynamics can be decomposed into 'purified' samples governed by one unique interaction law. RMSE measured between ground truth and GNN inferences is about $3 \cdot 10^{-2}$ ($14.3 \cdot 10^{6}$ positions, $\boldsymbol{x}_i \in [0, 1)^2$). The Sinkhorn divergence is about $7 \cdot 10^{-4}$.

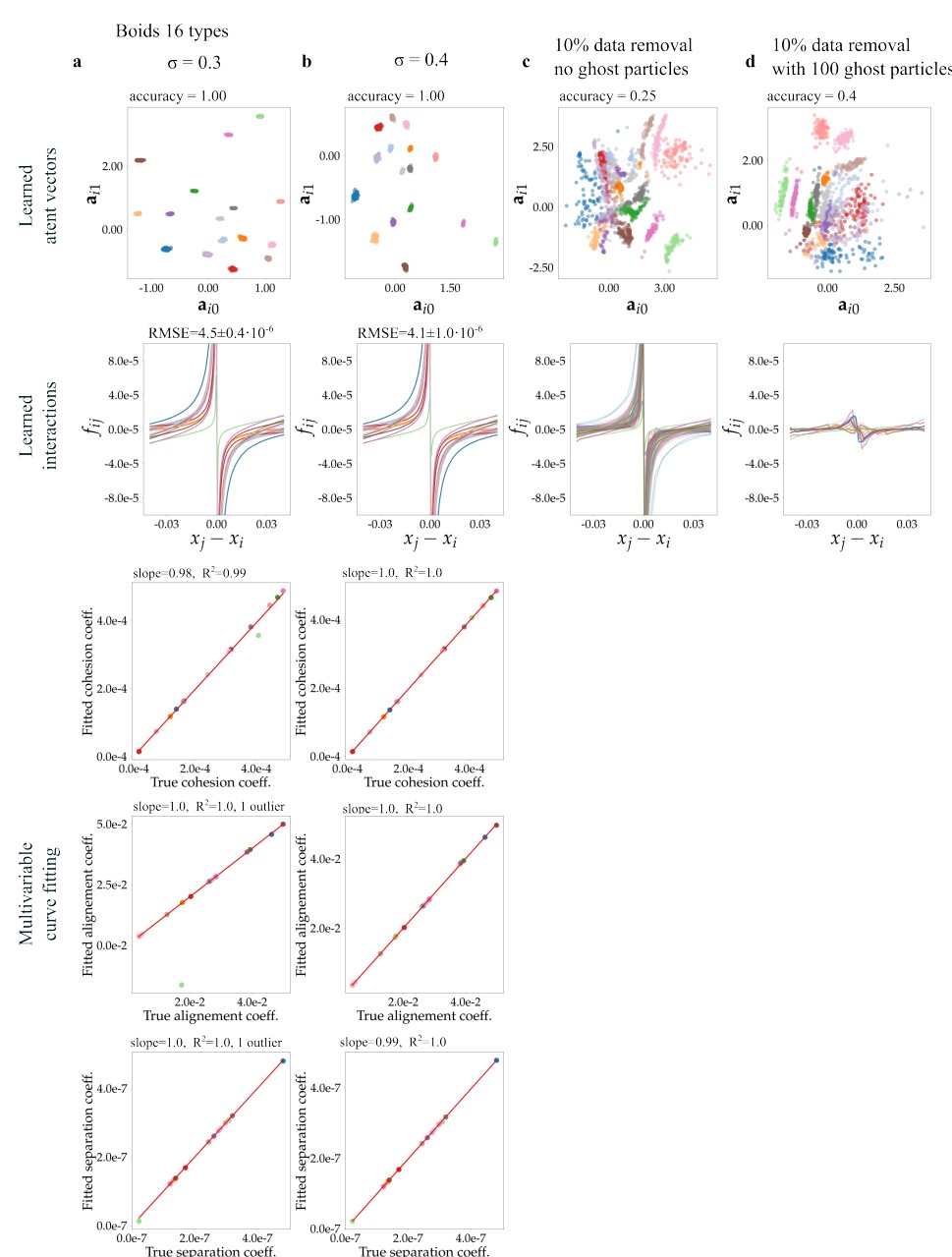

Supplementary Figure 14: Robustness to noise and data removal. Tests are performed with the GNN trained on a boids simulation (1,792 particles, 16 particle types, 8,000 time-steps). To corrupt the training with noise, we used a modified loss $L_{\ddot{x}} = \sum_i^n \|\hat{\ddot{x}}_i - \ddot{x}_i(1+\varepsilon)\|$ where $\varepsilon$ is a random vector drawn from a Gaussian distribution $\varepsilon \sim \mathcal{N}(0, \sigma^2)$. Results are shown for $\sigma = 0.3$ (**a**) and $\sigma = 0.3$ (**b**) . Row 1 shows the learned latent vectors $\boldsymbol{a}_i$ of all particles. Colors indicate the true particle type. Particles were classified with hierarchical clustering over the learned latent vectors $\boldsymbol{a}_i$. Row 2 shows a projection of the learned interaction functions $f$ as acceleration over distance for each particle. The last three rows show the results of supervised curve fitting of the interaction function $f$. Cohesion, alignment and separation parameters were extracted and compared to ground truth. (**c, d**) Randomly removing 10% of the training data yielded unsatisfying results that were not improved by adding ghost particles.

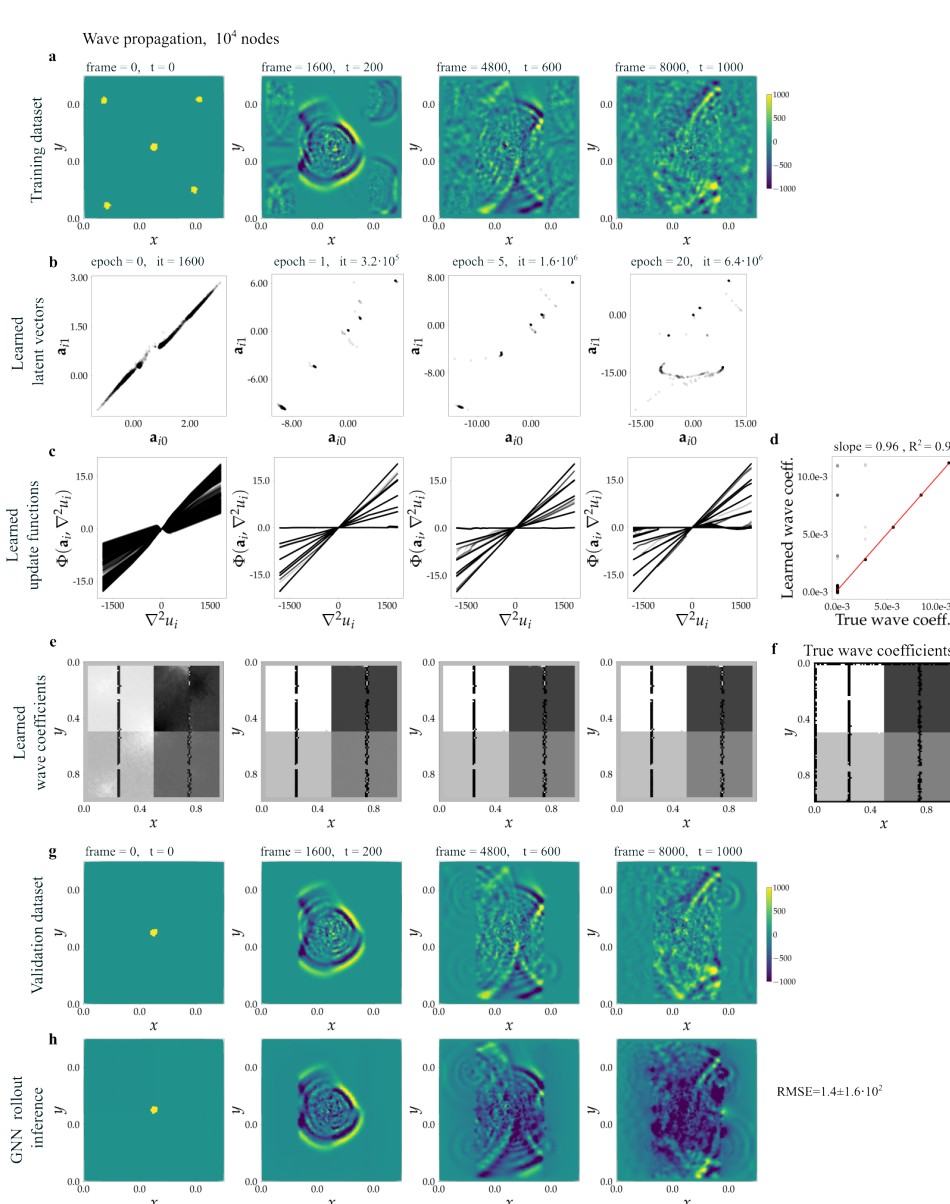

Supplementary Figure 15: GNN trained on a wave-propagation simulation. (**a**) The training dataset is a simulation of a scalar field $u_i$ evolving over a mesh of $10^4$ nodes with varying wave-propagation coefficients over space (**f**). Obstacles are modeled by particles with a wave-propagation coefficient of zero, there are two walls with four slits in the coefficient maps. (**b**) The learned latent vectors $\boldsymbol{a}_i$ are shown for a series of epochs and iterations. (**c**) The learned update functions $\Phi_i$ (Supplementary Table 2) over the discrete Laplacian of $u_i$ for all nodes $i$. Linear curve fitting of these profiles allows to extract the learned wave-propagation coefficients. (**d**) Comparison between true and learned wave-propagation coefficients. (**e**) The learned coefficients map are shown for a series of epochs and iterations. (**g**) Validation dataset with initial conditions different from training dataset. (**h**) Rollout inference of the trained GNN.

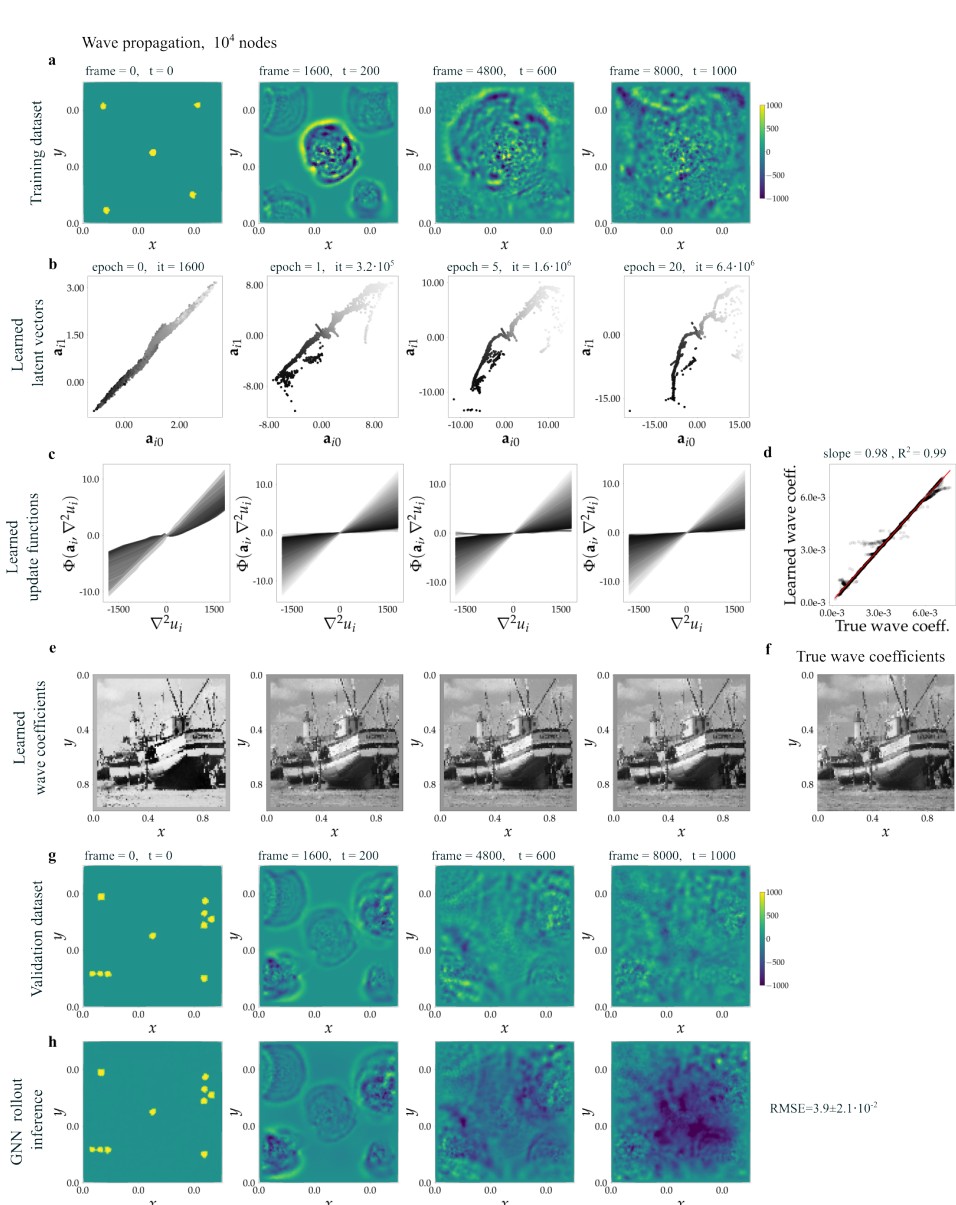

Supplementary Figure 16: GNN trained on a wave-propagation simulation. (**a**) The training dataset is a simulation of a scalar field $u_i$ evolving over a mesh of $10^4$ nodes with varying wave-propagation coefficients over space, here an arbitrary image (**f**). (**b**) The learned latent vectors $\boldsymbol{a}_i$ are shown for a series of epochs and iterations. (**c**) The learned update functions $\Phi_i$ (Supplementary Table 2) over the discrete Laplacian of $u_i$ for all nodes $i$. Linear curve fitting of these profiles allows to extract the learned wave-propagation coefficients. (**d**) Comparison between true and learned wave-propagation coefficients. (**e**) The learned coefficients map are shown for a series of epochs and iterations. (**g**) Validation dataset with initial conditions different from training dataset. (**h**) Rollout inference of the trained GNN.

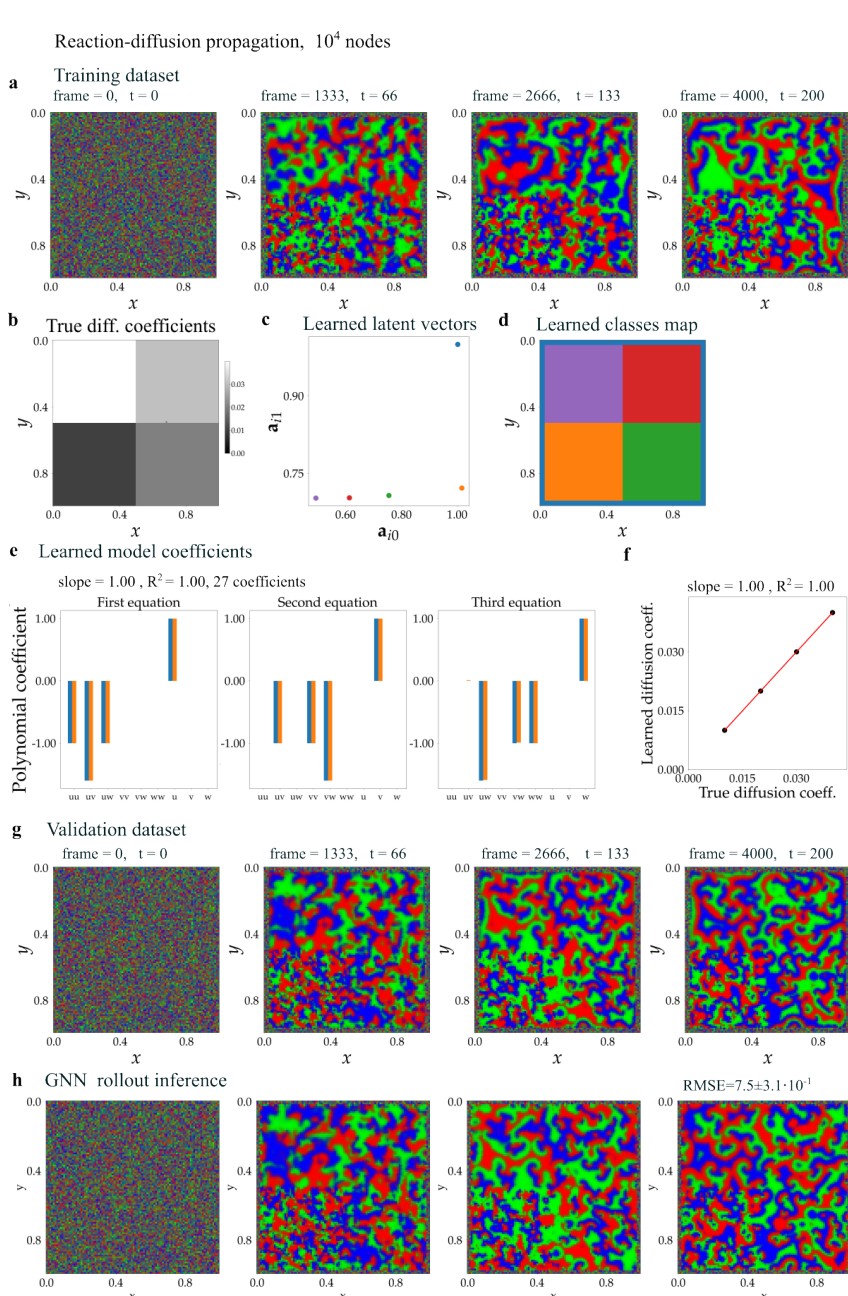

Supplementary Figure 17: GNN trained on a reaction-diffusion simulation based on the "Rock-Paper-Scissor" automaton. (**a**) The training dataset is a vector field $\{u_i, v_i, w_i\}$ evolving over a mesh of $10^4$ nodes (4,000 time-steps). The amplitudes of the field components are represented by red, blue, and green components respectively. (**d**) The diffusion coefficients vary over space. (**c**) The GNN learned that there were five distinct clusters in the latent vector embedding, including one for particles at the boundaries that follow a separate set of rules (**d**). We used all non-boundary particles to estimate the 4 diffusion coefficients and 27 polynomial function coefficients. (**e**) Comparison between true (blue) and learned (orange) polynomial coefficients. (**f**) Comparison between the true and learned diffusion coefficients. (**g**) Validation dataset with initial conditions different from training dataset. (**h**) Rollout inference of the trained GNN.

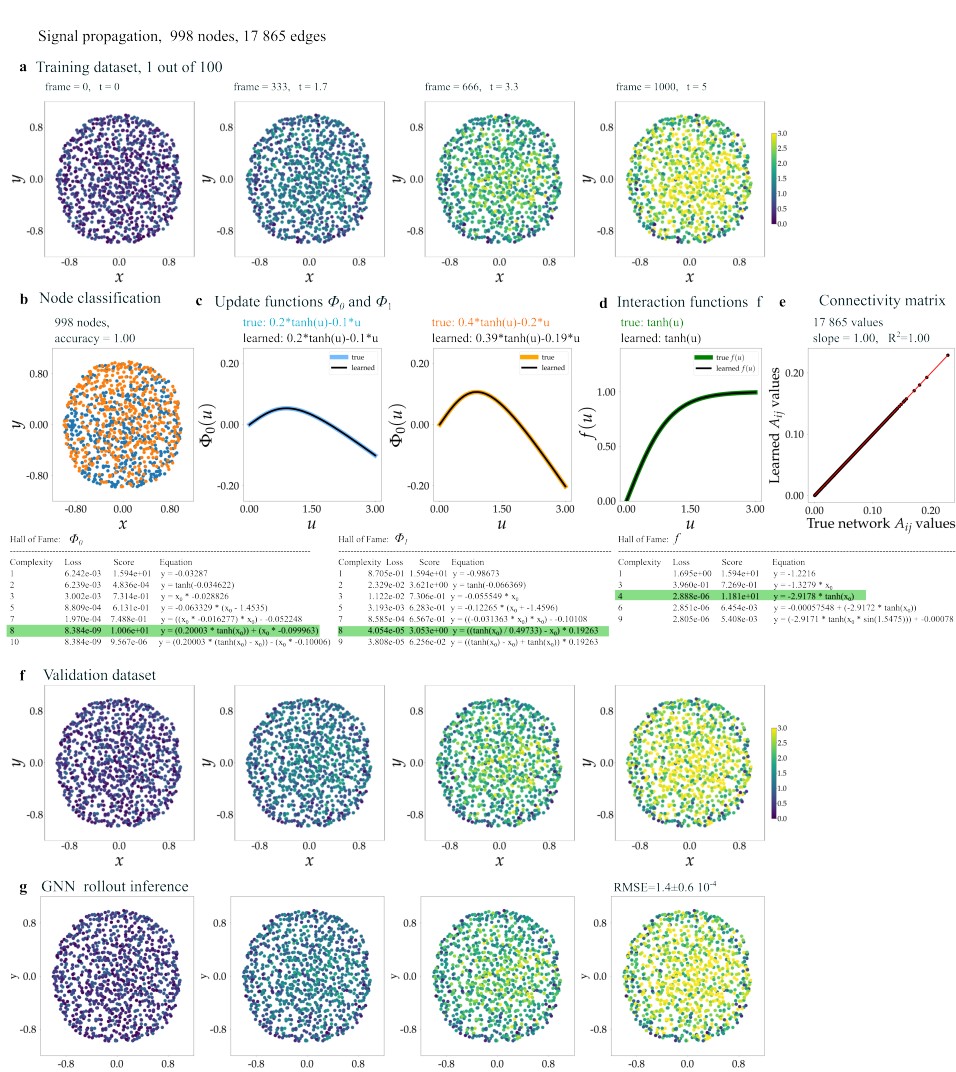

Supplementary Figure 18: GNN trained on a signaling network simulation. (**a**) The simulated network has 998 nodes connected by 17,865 edges. There are two types of nodes with distinct interaction functions. The training dataset consist of 100 simulations with different initial states run over 1,000 time-steps. (**b**) Threshold applied to the learned update functions Φ profiles shown in (**c**) allows to distinguish and classify the two node types. (**c**) Projection of the update functions Φ over node state $u$ and node types. Symbolic regression (PySR package) applied to the learned Φ functions allows to retrieve the exact expressions. (**d**) Projection of the learned interaction functions $f$ over node state $u$. This function is shared by all nodes and is well retrieved through the symbolic regression. The latter also retrieve a scaling scalar that is used to correct the learned connectivity matrix $\boldsymbol{A}_{ij}$ values. (**e**) Comparison between learned and true connectivity matrix values. (**f**) Validation dataset with initial conditions different from training dataset. (**g**) Rollout inference of the trained GNN over 1,000 time-steps.

## A.3 SUPPLEMENTARY VIDEOS

Hyperlinks removed, paper under double-blind review

**Video 1**: GNN trained on an attraction-repulsion simulation (4,800 particles, 3 particle types, 250 time-steps). Colors indicate the true particle type. (Left) Learned particle embedding training over 20 epochs. (Right) Projection of the learned interaction functions for each particle as speed over distance.

**Video 2**: GNN trained on an attraction-repulsion simulation. (Left) Validation dataset with initial conditions different from training dataset. Colors indicate the true particle type. (Right) Rollout inference of the fully trained GNN. Colors indicate the learned particle type.

**Video 3**: GNN trained on an attraction-repulsion simulation that is modulated by a hidden field of stationary particles playing a movie. (4,800 particles, 3 particle types, $10^4$ stationary movie particles). (Left) Particle embedding training over 20 epochs. Colors indicate the true particle type. (Middle) Projection of the learned interaction functions for each particle as speed over distance. (Left) frame 45 out of 250 of the learned hidden movie field.

**Video 4**: GNN trained on an attraction-repulsion simulation that is modulated by a hidden field of stationary particles playing a movie. (Left) True hidden movie field. (Right) learned hidden movie field. The grey levels indicate the coupling factors that modulate the interaction between the stationary particles and the moving particles.

**Video 5**: GNN trained on a gravity-like system (960 particles, 16 particle masses, 2,000 time-steps). (Left) Particle embedding training over 20 epochs. The colors indicate the true particle mass. (Right) Projection of the learned interaction functions for each particle as acceleration over distance.

**Video 6**: GNN trained on a gravity-like simulation. (Left) Validation dataset with different initial conditions than the training dataset. Colors indicate the true particle mass. (Right) Rollout inference of the trained GNN. Colors indicate the learned particle mass.

**Video 7**: GNN trained on a Coulomb-like simulation (960 particles, 3 particle charges, 2,000 time-steps). (Left) Validation dataset with different initial conditions than the training dataset. Colors indicate the true particle charge (ground truth). (Right) Rollout inference of the trained GNN. Colors indicate the learned particle charge.

**Video 8**: GNN trained on a boids simulation (1,792 particles, 16 particle types, 8,000 time-steps). (Left) Particle embedding training over 20 epochs. Colors indicate the true particle type. (Right) Projection of the learned interaction functions for each particle as acceleration over distance.

**Video 9**: GNN trained on a boids simulation. (Left) Validation dataset with different initial conditions than the training dataset. Colors indicate the true particle type. (Right) Rollout inference of the trained GNN. Colors indicate the learned particle type.

**Video 10**: GNN trained on a boids simulation (7,168 particles, 16 particle types, 8,000 time-steps). As a generalization test, the number of particles was multiplied by a factor of 4 (from 1,792 to 7,168) and arranged in as 16 homogeneous stripes. Colors indicate the true particle type. (Right) Rollout inference of the trained GNN. Colors indicate the learned particle type.

**Video 11**: GNN trained on a wave-propagation simulation over a field with varying wave-propagation coefficients. (Left) Particle embedding training over 20 epochs. Colors indicate the true wave-propagation coefficient. (Right) Learned wave-propagation coefficients.

**Video 12**: GNN trained on a wave-propagation simulation over a field with varying wave-propagation coefficients. (Left) Particle embedding training over 20 epochs. Colors indicate the true wave-propagation coefficient. (Right) Learned wave-propagation coefficients.

**Video 13**: GNN trained on a reaction-diffusion simulation based on the "Rock-Paper-Scissor" automaton over a mesh of $10^4$ 3D vector nodes (4,000 time-steps). The amplitudes of the vector

components are represented as RGB colors. Vector nodes have varying diffusion coefficients that modulate the reaction. (Left) Validation dataset with different initial conditions than the training dataset. (Right) Rollout inference of the trained GNN.

**Video 14**: GNN trained on a signaling propagation simulation. The training dataset has 998 nodes connected by 1,786 edges. There are two types of nodes with distinct interaction functions. (Left) Validation dataset with initial conditions different from the training dataset. Colors indicate signal intensity. (Right) Rollout inference of the trained GNN.

