# OpenReview forum: "Decomposing heterogeneous dynamical systems with graph neural networks"
_ICLR.cc/2025/Conference — Submitted to ICLR 2025_

### Official Review · Reviewer_KdqJ · 2024-10-31

**Soundness:** 3
**Presentation:** 3
**Contribution:** 2
**Rating:** 5
**Confidence:** 3

**Summary:**

This paper demonstrates that GNNs can be designed to jointly learn both interaction rules and heterogeneous structures directly from data. The learned latent structures and dynamics can then be used to decompose complex dynamic systems and infer the underlying governing equations. To evaluate the proposed approach, simulation experiments on moving particles and vector fields are conducted, highlighting its potential for capturing intricate dynamics.

**Strengths:**

The paper introduces a novel approach where GNNs jointly learn interaction rules and heterogeneous structures directly from data, enabling the decomposition of complex dynamic systems and inference of underlying governing equations. Simulation experiments on moving particles and vector fields demonstrate the model's effectiveness in representing complex dynamics, with promising potential for broader applications across various types of dynamic systems.

**Weaknesses:**

Novelty: Ordinary Differential Equations (ODEs) and Partial Differential Equations (PDEs) are well-established frameworks for modeling the evolution of dynamical systems over time. Several ODE/FDE-based GNNs, such as Continuous Graph Neural Networks (CGNN), have leveraged this approach to simulate time-evolving, interacting components in these systems. While the authors mention using simple MLPs instead of differentiable functions, it remains unclear what advantages MLPs offer over ODE-based methods. A discussion on this aspect would clarify the benefits of MLPs in the proposed approach and strengthen the novelty claim. Is it possible to provide a comparative analysis between their MLP-based approach and ODE-based methods like CGNNs, and also discuss the relative advantages and disadvantages in terms of computational efficiency, expressiveness, or ease of implementation?

Lack of experiments over real data: While the approach is tested on simulated data, it lacks validation on real-world data observed in natural systems. This limitation raises concerns about the method’s practical effectiveness and generalizability to real-world dynamics, where noise and complexities may differ significantly from simulations. Including experiments with real data would strengthen the evidence for the approach’s applicability, for example, applying the proposed method to data from physical experiments, biological systems, or social networks is more convincing.

**Questions:**

Please see the weakness.

---

> ### Author Response · Authors · 2024-11-25
>
> We are not aware of an ODE/PDE or GNN-based ODE/PDE simulator that infers a heterogeneous set of latent parameters from observed dynamics in a way comparable to our method.
>
> We used MLPs not because we believe that they are a superior architecture, but because they are the simplest method that generated excellent outputs and allowed us to infer the structure of the latent parameterization in a way that supports further analysis like inferring and fitting symbolic functions.
>
> We agree that experiments with real data would be great, but beyond the scope of this manuscript.  This line of future work is already discussed in the manuscript.

---

> ### Comment · Reviewer_KdqJ · 2024-11-26
>
> There are multiple ODE/FDE based neural networks, for example, [1-2], and those papers describe how features associated with graph neural network are evolved with time. I do not aware why an ODE/PDE or GNN-based ODE/PDE simulator could not be compared with your method. Maybe I miss something. As the author has not addressed my concern, I have decided to retain my initial score.
>
> [1] Unleashing the Potential of Fractional Calculus in Graph Neural Networks with FROND.
> [2] Graph Neural Convection-Diffusion with Heterophily.

---

> > ### Author Response · Authors · 2024-11-27
> >
> > Dear reviewer,
> >
> > Thanks a lot for letting us know about the two references on ODE-GNNs that you were thinking of. Both manuscripts address diffusion networks without rewiring which is related to our experiments with wave propagation, reaction diffusion, and signaling
> > networks.  [1] addresses static graphs and contributes fractional derivatives, which is interesting to implement memory, but only
> > tangentially related to our current work.  We think we can consider this mechanisms in future version that deal with this problem space, thanks for the pointer.  [2] also looks very interesting.  They jointly solve for normal 'homophilic' diffusion and 'heterophilic' convection using different learnables.  Interestingly, the inputs to the 'heterophilic' term is the difference of the observable feature vectors of two nodes, that is, only the difference matters, not the node type itself.  Compared to not considering heterogeneity at all, this approach improves performance in a number of node classification tasks, which makes a lot of sense.  However, we think that it is significantly different from our aims, such that generating a comparison would require (a) the generation of different datasets, (b) deviating far from our objective (identifying hidden parameters, interaction- and update functions).  Our manuscript focuses on how to utilize learned embeddings to infer the governing rules underlying the dynamics, [2] stops at demonstrating that considering heterogeneity improves the predictive (classification) performance of GNNs in established benchmarks.
> >
> > Since our manuscript is not focused on building systems that improve predictive performance, we do not believe that conducting such a contrived comparison provides additional value.  Also, the method is not designed to address dynamical systems that rewire over time, and to infer connectivity and complex interaction functions as in the signaling network example.
> >
> > We are adjusting our introdution to address this work, because it expands ODE/PDE-GNNs to learn diffusion like system by a notion of heterogeneity that we missed before.  Thanks again for your help!

---

### Official Review · Reviewer_ca5v · 2024-10-31

**Soundness:** 2
**Presentation:** 2
**Contribution:** 3
**Rating:** 3
**Confidence:** 2

**Summary:**

This paper demonstrated that a graph neural network can both learn the dynamics and structure of a dynamical system. The parameters showed both underlying structure and differential equations. The proposed model was tested on several simulated datasets, and visually showed a meaningful relationship between the true and predicted values.

**Strengths:**

This model can be applied to various complex realistic systems to reveal the underlying structure and dynamics.

**Weaknesses:**

1. The presentation is poor. It is hard to track simulations they did for the model. People need to check figure, table, video and supplementary figure to understand what they have done without any hint or explanation.
2. The interpretations are lack to explain their results. Only showing several latent representations are not enough to convince or help understand the model's strength.

**Questions:**

1. Have you compared your proposed model with a conventional GNN, i.e., the loss is l2-norm between x and x_hat? What is the significant improvement by the proposed model?
2. How do you choose to use first-order or second-order derivative to train the model?
3. At a single time point, why don't you use both first-order and second-order or even with higher-order derivative to train the model, where the information should be more compact?

---

> ### Author Response · Authors · 2024-11-25
>
> We take the comment about presentation to heart and will improve the text.  We show that our method inferred the heterogeneous set of latent parameters, unknown interaction and update functions, and achieved excellent rollout performance.  We also showed that the learned functions, sampled from the learned embedding space, can be used as sample generators to infer symbolic interaction functions and their parameters.  This is in fact already in the text, but we will adjust the manuscript to make this point more comprehensible.
>
> Q: Have you compared your proposed model with a conventional GNN, i.e., the loss is l2-norm between x and x hat? What is the significant improvement by the proposed model?
>
> A: We use a conventional message passing GNN with the addition that the learnable components are parameterized by observable state AND a learnable latent to infer the structure of the heterogeneity.  This is new and so there is no meaningful baseline to compare to.
>
> Q: How do you choose to use first-order or second-order derivative to train the model?
>
> A: We chose experiments for either to demonstrate that either is possible.  In a real world experiment we would likely experiment with both.
>
> Q: At a single time point, why don't you use both first-order and second-order or even with higher-order derivative to train the model, where the information should be more compact?
>
> A: We did this and it works, but it did not contribute to the clarity of the experiments.
> In the Boids experiment, we use both first and second order derivatives.  In the Gravity experiment, we added the second derivative even though it is not present in the real equation. As desired, the GNN correctly learned to ignore this information.  We will improve the text to make this clearer.

---

### Official Review · Reviewer_LaeA · 2024-11-04

**Soundness:** 1
**Presentation:** 1
**Contribution:** 1
**Rating:** 3
**Confidence:** 5

**Summary:**

This paper fits GNNs to dynamic systems and showed that it is feasible.

**Strengths:**

- The figures are well-presented.
- The experiment details are clear.

**Weaknesses:**

- First of all, what does heterogeneity mean in this context? Meaning that the particles are of different types? The closest thing I can find to a definition is "The latent heterogeneity of the particles is encoded by a two-dimensional learnable embedding $a_i$ that is part of the node features." But why two?
- The simulation of dynamic systems has been routinely done by GNNs, most notably Sanchez-Gonzalez et al., 2020, as cited in the paper. The only difference seems to be that they do not consider this "heterogeneity" whose definition is not clear? But does this method introduced here even outperform the baselines without such input?
- This paper uses a very classical method that is not equivariant on 3-dimensional space. How does it deal with rotational equivariance?
- There is no baseline nor quantitative comparison.
- No code is provided.

**Questions:**

See Weaknesses.

---

> ### Author Response · Authors · 2024-11-25
>
> Heterogeneity is the opposite of homogeneity, meaning that things are different and not same.  The way in which they are different is often not known when we observe natural phenomena.  It can be cell types, mass, age, ... discrete or continuous latent parameters of arbitrary dimensionality that control unknown aspects of unknown rules underlying the observable dynamics.  As there are a lot of unknowns, we were looking for a system that allows us to control some aspects of the dynamics while learning others, in an interpretable and practically useful way to infer those rules underlying the dynamics.  We all know that GNNs have been shown to be an excellent tool to model such dynamical systems, including interactions between heterogeneous elements.  To our knowledge, there was only one attempt though to learn such latent variables together with the interaction laws from data: the orbital mechanics work by Lemos et al. (2023). However, they chose to explicitly learn a latent scalar $b$ that the learnable function approximator $F$ does not have access to $b F(x)$. In contrast, we learn a latent vector $a$ that the learnable function can use to model heterogeneities among the parts of the dynamical system: $F(a,x)$.  We showed that this allows us to deal with heterogeneous behavior caused by 1 to 4-dimensional latent parameters regardless of the dimensionality of $a$.  To our knowledge, this is new, so there is no meaningful baseline method to compare to.  E.g., if we train LG-ODE with different spring constants or Learning-to-simulate without correct particle types, we do not get meaningful results, because those methods are not designed to infer these latent parameters.
>
> For the dimensionality of the latent space we chose 2 in our experiments, because with 1, the network got stuck in local minima (it is hard to walk around obstacles on a line), and 3 or more did not improve results, regardless whether the underlying latent parameter space was 1, 2, 3 or 4D.  A 2D embedding space is also easy to visualize and helps human interpreters to analyze the result, which is great.  We will improve the text to address this point.
>
> We achieve effective rotational invariance by augmenting training samples by random rotation where appropriate (i.e., not in signaling networks).  The simulated examples are 2D quasi-physical systems, but the approach works similarly for other dimensionalities.
>
> The final paper will include a link to our GitHub repository with code for all experiments under a permissive open source license (we omitted including the link in the first submission as this would have compromised double blind review).

---

> > ### Comment · Reviewer_LaeA · 2024-11-27
> >
> > Thank you for your rebuttal.

---

### Official Review · Reviewer_tcQf · 2024-11-12

**Soundness:** 3
**Presentation:** 3
**Contribution:** 2
**Rating:** 5
**Confidence:** 4

**Summary:**

The paper proposes using graph neural networks (GNNs) to jointly learn interaction rules and heterogeneous structure in complex dynamical systems from data alone. Extensive experiments on simulated systems including particle interactions, wave propagation, reaction-diffusion, and signaling networks, showing its good performance.

**Strengths:**

1. The paper is in general easy to follow and with clear writing flow. The problem is well-motivated, by using GNN to learn system dynamics over time and in the meanwhile, uncover the underlying latent properties in an interpretable way that facilitates further analysis.

2. The evaluation of dynamical systems in the experiment sections are extensive, though adding some baselines for comparison would be better.

**Weaknesses:**

1. There is no related work section. Some works are discussed in the introduction part, but there are many existing neural simulators that use GNN to rollout trajectories of multi-agent dynamical systems [1,2,3,4]. Discussion about existing work and comparison in the experiment section are helpful to provide a comprehensive analysis.

2. As mentioned above, for rollout MSE across different datasets, it is suggested to compare against representative baselines. Also the run time comparison can be included across compared methods.





[1]  Learning Continuous System Dynamics from Irregularly-Sampled Partial Observations.

[2] Interaction Networks for Learning about Objects, Relations and Physics.

[3] Learning to simulate complex physics with graph networks.

[4] HOPE: High-order Graph ODE For Modeling Interacting Dynamics

**Questions:**

The authors mention the method can infer the underlying governing equations, (line 17) but I do not see any analysis in the experiment part. It would be interesting to see how can we extract formula from a learned GNN.

---

> ### Author Response · Authors · 2024-11-25
>
> We discuss related work in the introduction, including the suggested [3] work by Sanchez-Gonzales et al. (2020), which is not designed to learn latent properties (different materials or parameters).  The follow-up work by Lemos et al. (2023) is the most related to ours and is therefore discussed most extensively. They learn one latent property and one unknown interaction law.
>
> Beyond what Lemos et al. (2023) demonstrated, we infer discrete and continuous latent parameters between 1 and 4 dimensions that vary across particles, a variety of diverse interaction functions, external inputs, and connectivity matrices.
>
> Instead of the suggested reference [2], we cited Gilmer et al. (2017) which we found to cover the same conceptual ideas, we will add [2] in the same context.
>
> Suggested references [1] and [4] are similar in scope, yet do not explicitly address learning a latent parameterization to infer the structure of heterogeneous interaction laws.  Since open source code is available for [1] (LG-ODE), but not for [4], we conducted some experiments with this code base.  Briefly, [1] trains a GNN-ODE-VAE architecture to inpaint trajectories of particles that are connected by zero-length springs.  The training data is of considerable size, 2500 simulations of ~100 timesteps each, 2000 for training and 500 for testing, each containing 5 particles of which some are connected by springs.   The connectivity is provided during training and inference, the spring constant is 1 for all springs, and the pairwise interaction law is $F = -x$.  The method's goal is solely to predict particle positions over time, and it is not meant to infer the connectivity matrix or diverse spring constants, nor to provide insights into the structure of the dynamical system.  It is therefore not directly comparable to ours.  We successfully trained our networks to infer diverse spring constants, connectivities from similar training data and achieved excellent inpainting and rollout performance.  Since those experiments are a significant addition, and since the experiment does not contribute meaningfully to the manuscript, we would prefer to not add them to the supplement.
>
> Q: The authors mention the method can infer the underlying governing equations, (line 17) but I do not see any analysis in the experiment part. It would be interesting to see how can we extract formula from a learned GNN.
>
> A: Similar to Lemos et al. (2023), we used Symbolic Regression (PySR) to infer appropriate symbolic functions and their parameters by using the trained network as sample generators.  PySR retrieved the correct function for gravity, charged particles, and signaling networks, but failed for the Boids and RPS experiments.  Once a symbolic function is established, fitting parameters this way worked for all experiments.

---

### Author Response · Authors · 2024-11-25
**Working on improving the text**

Dear reviewers, thanks a lot for your candid reviews.

We realize that we failed to make it clear how our approach is different from the related work that you mention and that we discuss in the introduction.  We are in the process of changing the text to make this more clear, but in the meantime, here is a short summary of what's new:

1. Simulation: We simultaneously learn unknown interaction- and update functions and an embedding of latent heterogeneous properties from observations of dynamical systems.
2. Interpretation: We do this in a way that makes it easy to use the learned embedding and functions to infer the underlying rules and properties governing the dynamical system.

We achieve this by training simple GNNs with single MLPs for interactions and/or updates of particle states that are parameterized by the observable states and a low dimensional learnable latent vector for each particle.  We then sample the learned functions from the learned latent embedding space and infer symbolic rules and their parameters, similar to the work on orbital dynamics by Lemos et al (2023).  The significant difference to their approach is that they chose to explicitly learn a latent scalar $b$ corresponding to a latent factor multiplied with a learnable function $b F(x)$.  In contrast, we learn a latent vector $a$ that the learnable function $F(a,x)$ uses to map out an arbitrary latent parameter space.  We chose the latent vector $a$ to be 2-dimensional which was the lowest dimensionality that worked in all our experiments (it also prints well on paper and is easy to interpret).

We kept the examples as simple as possible to demonstrate how this can be used for a diverse set of simulated quasi-physical simulations whose interactions depend on 1 to 4-dimensional latent parameters, and show how the learned MLP and the embedding can be used to infer symbolic interaction functions and their parameters.

To demonstrate how this approach can be extended for more complex systems, we added an example that includes latent external inputs that affect---but themselves are not altered by---the dynamical system (Fig. 4), and an example where the interaction matrix has to be inferred (Suppl. Fig. 18).

To our knowledge, this approach is new, so there is no meaningful baseline to compare to.  E.g., if we train LG-ODE with different spring constants or Learning-to-simulate without known particle types, we do not get meaningful results, because those methods are not designed to infer these latent parameters.

Even though we were not successful at generating reasonable results from training data with variable spring constants with LG-ODE, we think that it should be possible for methods without a dedicated latent state embedding to infer plausible behavior from extended temporal context if sufficient training data is provided (e.g., LG-ODE uses the entire time series, whereas we use a single time step).  However, the underlying properties responsible for this aspect of the behavior (e.g., variable spring constants) would be significantly more difficult to extract from the trained parameters of the network.  Interestingly, one could likely use our method to infer them from the learned simulation.
We think that those topics would be super interesting to investigate in future work but are not within the scope of this manuscript.

We hope that you agree with us that our presented method is an exciting new way to use GNNs to infer the rules and latent properties of physical and biological dynamical systems and we would love to discuss future experiments and ideas with conference participants at ICLR.

---

### Meta-Review · Area_Chair_2NPm · 2024-12-17

**Metareview:**

The paper proposes a graph neural network framework for learning the dynamics of a physical system, which is tested on several simulations.

We had four reviews, all negative. All reviewers are unanimous concerning the lack of novelty, poor writing and presentation (e.g., Figure 1 in the paper), lack of real-world simulations, missing baselines, and several other concerns.

I see no reason to overrule this consensus, and I suggest to reject the paper.

**Additional Comments On Reviewer Discussion:**

- **Reviewer KdqJ** was concerned about the novelty of the paper and the lack of experiments on real-world datasets. The rebuttal did not address these points, and the reviewer remained negative.

- **Reviewer ca5v** highlighted the poor presentation / writing of the paper, and the fact that the experimental validation was insufficient. There was no time in the rebuttal to discuss these points.

- **Reviewer LaeA** provided a short review with some concerns on the novelty, the results, and the presentation.

- **Reviewer tcQf** was concerned about the lack of a related works section and some missing baselines.

Overall, all concerns are valid, and I considered all these points in my final evaluation.

---

### Decision · Program_Chairs · 2025-01-22

Reject